

# The sensitivity of estuarine aragonite saturation state and pH to the carbonate chemistry of a freshet-dominated river

Benjamin L. Moore-Maley[1], Debby Ianson[1,2], and Susan E. Allen[1]

[1]Department of Earth, Ocean and Atmospheric Sciences, University of British Columbia, Vancouver, British Columbia, Canada
[2]Fisheries and Oceans Canada, Institute of Ocean Sciences, Sidney, British Columbia, Canada

*Correspondence to:* Benjamin Moore-Maley (bmoorema@eoas.ubc.ca)

**Abstract.**

Ocean acidification threatens to reduce pH and aragonite saturation state ($\Omega_A$) in estuaries, potentially damaging their ecosystems. However, the impact of highly variable river total alkalinity (TA) and dissolved inorganic carbon (DIC) on pH and $\Omega_A$ in these estuaries is unknown. We assess the sensitivity of estuarine surface pH and $\Omega_A$ to river chemistry using a 5 1-dimensional, biogeochemical-coupled model of the Strait of Georgia on the Canadian Pacific coast and generalize the results in the context of global rivers. The productive Strait of Georgia estuary has a large, seasonally variable freshwater input from the glacially fed, undammed Fraser River. Analyzing TA and pH observations from this river and its estuary, we find that the Fraser is moderately alkaline (TA 500–1350 $\mu$mol kg$^{-1}$) but relatively DIC-rich, especially during winter (low flow). Model results show that estuarine pH and $\Omega_A$, while sensitive to freshwater DIC and TA, do not vary in synchrony. Instead, rivers with 10 high DIC and TA produce lower estuarine pH due to an increased estuarine DIC:TA ratio, but higher estuarine $\Omega_A$ because of DIC contributions to the carbonate ion. This estuarine pH sensitivity decreases with increasing mean river TA, but the zone of maximum pH sensitivity also moves to higher salinity which could impact a larger areal extent of the estuary. Many temperate rivers, such as the Fraser, are expected to experience weaker freshets and stronger winter flows under climate change, reducing the extent of the river plume and the impact of river chemistry in much of the estuary. However, increasing carbon in rivers 15 will move the highest sensitivity zone to higher salinities that cover larger areas under present-day flow regimes.

## 1 Introduction

Estuaries support productive ecosystems (Cloern et al., 2014) and significant human populations (Cloern et al., 2015). Critical trophic links within many of these ecosystems may be negatively impacted by increases in dissolved inorganic carbon (DIC) and reduced pH associated with ocean acidification (Haigh et al., 2015). Those organisms using the calcium-carbonate mineral 20 form aragonite in their external hard parts (e.g., mussels, oysters, geoduck) are especially vulnerable since oceanic $CO_2$ uptake lowers the aragonite saturation state ($\Omega_A$) of seawater (Waldbusser et al., 2015). While carbonate system dynamics have been studied extensively in marine (e.g., Jiang et al., 2015) and freshwater (e.g., Schindler, 1988) environments, less is known about carbon chemistry in the estuaries where these two zones meet (Salisbury et al., 2008).



Estuarine systems are complex, cover large salinity ranges that challenge current measurement techniques and are generally under-sampled (Ianson et al., 2016). The carbonate chemistry in the rivers that feed these estuaries may also be exceptionally variable, ranging from rivers with low DIC to total alkalinity (DIC:TA) ratios and high pH ($> 8$) like the Mississippi River (Hu and Cai, 2013) to blackwater rivers that have low pH ($< 5$) (Cai et al., 1998; de Fátima F. L. Rasera et al., 2013).

Along a salinity gradient there may exist maximum sensitivity zones occurring where DIC$\sim$TA (Egleston et al., 2010) that are especially vulnerable to acidification. The strength and location of these zones depend on the river endmember carbonate chemistry (Hofmann et al., 2009; Hu and Cai, 2013). In addition, seasonality is often strong and single rivers and estuaries may experience highly variable conditions (Waldbusser and Salisbury, 2014; Hunt et al., 2014; Ianson et al., 2016). On top of natural variability, many estuaries also experience heavy anthropogenic pressure (Frankignoulle et al., 1996; Zhai et al.,

2007), making them particularly vulnerable to ocean acidification (Cai et al., 2011) and in some cases the subject of intensive management and policy initiatives (Fennel et al., 2013).

DIC and TA vary by nearly two orders of magnitude between major world rivers (Cai et al., 1998), and throughout individual watersheds in space and time (Hellings et al., 2001; de Fátima F. L. Rasera et al., 2013; Voss et al., 2014). Carbonate and silicate weathering are major sources of river DIC and TA in most rivers (Meybeck, 1987; Amiotte Suchet et al., 2003). Both

quantities can be strongly flow-dependent due to dilution if physical weathering outpaces chemical weathering rates (i.e., kinetic-limited weathering (West et al., 2005)). Carbonates are concentrated globally in the northern mid-latitudes (Amiotte Suchet et al., 2003), and several carbonate-rich, mid-latitude watersheds demonstrate high TA and large TA flow-dependence (e.g., Changjiang, Mississippi (Cai et al., 1998)). TA flow-dependence is also common among low-TA, low-latitude rivers (e.g., Amazon (Richey et al., 1990), Congo (Wang et al., 2013)), although organic carbon contributions can complicate this behavior.

In the Congo River for example, TA is flow-dependent but DIC is persistently high year-round (Wang et al., 2013). Pollution can also contribute enormously to TA and DIC throughout urbanized watersheds (e.g., Scheldt (Hellings et al., 2001)) and estuaries (e.g., Changjiang (Zhai et al., 2007)).

In the present study, we determine the sensitivity of $\Omega_A$ and pH in a large, mid-latitude, fjord estuary (Strait of Georgia, Canada) to changes in freshwater TA and pH using a one-dimensional biogeochemical model. We establish freshwater TA

and pH ranges and variability for the system by extrapolating TA observations from the Fraser River plume region (de Mora, 1983; Ianson et al., 2016) to zero salinity, and by using autonomous pH measurements from an Environment and Climate Change Canada (ECCC) mooring near the Fraser River mouth. We use TA and pH to constrain freshwater DIC since we lack reliable DIC observations in the Fraser River. From the model $\Omega_A$ and pH results across 18 scenarios based on these estimated freshwater TA and pH ranges, we identify regions of the freshwater pH (DIC:TA) range that produce enhanced estuarine pH and

$\Omega_A$ sensitivity to freshwater TA. We characterize these regions in terms of past (higher pH) and future (lower pH) freshwater carbonate chemistry. We further discuss the implications of these results for global rivers and future climate.



## 2   Methods

### 2.1   Study area

The Strait of Georgia (Fig. 1) is a large ($\sim$6800 km$^2$, >400 m deep), temperate, semi-enclosed, fjord-like estuarine sea with strong seasonal stratification, productivity, and carbonate chemistry cycles (Moore-Maley et al., 2016; Ianson et al., 2016). This

high productivity supports abundant populations of shellfish, finfish, and other higher organisms that may be sensitive to pH and $\Omega_A$ anomalies (Haigh et al., 2015). The Fraser River, the primary freshwater source, drains approximately 238,000 km$^2$ with seasonally-variable discharge ($\sim$800 to 12,000 m$^3$ s$^{-1}$ at Hope, ECCC data, http://wateroffice.ec.gc.ca) due to summer snow/ice melt and lack of dams throughout most of the watershed. This large freshwater flux is partially contained by narrow passages and tidal mixing over sills (Pawlowicz et al., 2007), and thus imparts a significant freshwater influence on the Strait

especially compared with regions where large rivers meet the ocean directly such as the nearby Columbia River plume region (Roegner et al., 2011). These same coastal and topographic features create long residence times, causing carbon to accumulate and making the Strait DIC-rich relative to the open ocean (Ianson et al., 2016) despite a strong, seasonal DIC upwelling signal over the outer shelf (Bianucci et al., 2011). The large freshwater footprint, together with the abundance of previous circulation (e.g., LeBlond, 1983; Pawlowicz et al., 2007), ecology (e.g., Masson and Peña, 2009; Allen and Wolfe, 2013), and acidification

studies (e.g., Moore-Maley et al., 2016; Ianson et al., 2016) make the Strait an ideal system for investigating the response of estuarine pH and $\Omega_A$ to freshwater carbonate chemistry.

The Fraser River watershed spans four distinct geologic belts (Fig. 1a) that transition from the carbonate-rich Foreland Belt to the silicate-rich Coast Belt (Cameron and Hattori, 1997). Carbonate and silicate weathering thus dominate the watershed (Voss et al., 2014); carbonate weathering generally produces TA faster than silicate weathering (Meybeck, 1987). Observed

Fraser River TA (Jul/Aug 2009, Oct 2010, May/Jun 2011, (Voss et al., 2014)) accumulates along the Foreland Belt, decreases along the Coast Belt, and is highest at low flow stage. Fraser River TA thus appears to be produced primarily by carbonate weathering in the upper watershed, diluted by weakly-buffered seaward tributaries, and flow-dependent. Remineralization of dissolved organic matter can contribute to DIC (e.g., Yukon (Wickland et al., 2012)), and high river turbidity can limit biological DIC uptake (Irigoien and Castel, 1997) as is the case in the Fraser River (Moore-Maley et al., 2016). Like most rivers, the Fraser

is persistently aragonite undersaturated ($\Omega_A < 0.3$). However, some alkaline (TA > 2000 $\mu$eq kg$^{-1}$), mid-latitude rivers have been estimated to be near or supersaturated with respect to aragonite (Salisbury et al., 2008).

### 2.2   Data sources

TA can behave, at times, like a conservative tracer in the Canadian coastal Pacific Ocean (Ianson et al., 2003). We therefore estimate the Fraser-dominated, seasonal, freshwater TA endmembers for the Strait of Georgia by extrapolating TA observations

from 10 sampling cruises near the Fraser plume (de Mora (1983); Ianson et al. (2016); Table 1) to zero salinity ($S = 0$) using linear regression (Fig. 2). We only consider profiles or transects that include at least one TA sample below $S = 20$ to ensure sufficient river influence, which limits our analysis to 5 cruises prior to 1980 (March, May, October 1978 and January, April 1979; de Mora (1983)) and 5 cruises after 2010 (August, October 2010, June, August 2011 and July 2012; Ianson et al. (2016)).





Unlike TA, DIC does not behave conservatively in the Strait of Georgia. We therefore calculate the freshwater DIC endmember from the extrapolated TA endmembers and a Fraser River pH range determined using observations from the ECCC Fraser River Water Quality Buoy (Table 1) moored approximately 10 km upstream along the main arm of the river. Buoy pH was measured potentiometrically using a regularly-inspected (bimonthly to monthly), hull-mounted, YSI ADV6600 multisensor and recorded hourly on the National Institute of Standards and Technology (NIST) scale between 2008 and 2013 (Ethier and Bedard, 2007). We calculate DIC from TA and pH using the CO2SYS program (Lewis and Wallace, 1998) and full salinity range $K_1$ and $K_2$ constants (Millero, 2010).

The strong likelihood of non-conservative TA behavior at low salinity, organic alkalinity contributions, and poor accuracy of potentiometric pH measurements introduce uncertainty into our estimates of freshwater TA, pH, and DIC. Non-conservative TA behavior has been observed in the Fraser plume at low salinity (de Mora, 1983), particularly during the October 1978 cruise where TA near $S = 0$ varied between approximately 700 and 1200 $\mu$mol kg$^{-1}$. This low salinity variability may be associated with sedimentation of charged clay particles (Kennedy, 1965). If organic acids and bases contribute significantly to TA, as they do in some coastal areas (Koeve and Oschlies, 2012; Kim and Lee, 2009; Hernández-Ayón et al., 2007) and rivers (Hunt et al., 2011; Kennedy, 1965), carbonate alkalinity estimates (which are required to calculate DIC) become increasingly inaccurate. Carbonate alkalinity is estimated to be as low as 10% of TA in the Congo (Wang et al., 2013) and Kennebec Rivers (Hunt et al., 2014). The carbonate alkalinity fractions of TA in the Fraser River and Strait of Georgia are unknown. Potentiometric pH measurements in seawater are generally no more precise than 0.02 units (Byrne et al., 1988; Dickson, 1993), however inconsistencies in electrode type and calibration can produce errors in freshwater larger than 0.1 units (Covington et al., 1983).

Given these uncertainties, we consider the extrapolated TA endmembers, the buoy pH record, and the resulting calculated freshwater DIC to represent approximate seasonal ranges of freshwater carbonate chemistry rather than absolute values. To provide context to these TA ranges, we compare our endmembers to TA data from ECCC Fraser River sampling programs near the ECCC Water Quality Buoy and approximately 100 km upstream near the town of Hope, BC (Fig. 1; Table 1). Since neither dataset was sampled to oceanographic standards – specifically samples were unpreserved and stored in polyethylene containers – we use these data for comparison only and not to define our freshwater carbonate chemistry endmembers.

## 2.3 Model

### 2.3.1 Overview

We employ a one-dimensional (1-D), biogeochemical coupled, Strait of Georgia mixing model that resolves the upper 40 m of the water column on a 15 minute timestep and predicts annual cycles (Moore-Maley et al., 2016) in order to investigate the sensitivity of surface estuarine pH and $\Omega_A$ in the Strait to changes in river carbonate chemistry. Three-dimensional estuarine circulation is parametrized as an upward entrainment velocity and outward advective flux, both defined in terms of the total freshwater discharge from the Fraser River and other small rivers (Collins et al., 2009). The advective loss arises due to water column convergence, since the upward entrainment velocity increases with depth. We explicitly model *in situ* temperature (ITS-90 (Preston-Thomas, 1990)) and practical salinity (PSS-78 (UNESCO, 1981)) as physical state variables.



The biological model contains 3 nutrient classes (nitrate, ammonium, dissolved silica), 3 photosynthesizer classes (diatoms, heterotrophs as *Mesodinium rubrum*, nanoflagellates), 3 grazer classes (*M. rubrum*, microzooplankton, mesozooplankton), and 3 detritus classes (dissolved and particulate organic nitrogen, biogenic silica). *M. rubrum* (a ciliated protozoan) retains functional chloroplasts during grazing and uses them to perform photosynthesis. Calcifying phytoplankton (e.g., coccolithophores) are assumed to contribute minimally to productivity in the Strait of Georgia (Haigh et al., 2015) and were absent from satellite observations in the Strait prior to 2016 (J. Gower, personal communication, 2014; NASA Earth Observatory, http://earthobservatory.nasa.gov/IOTD/view.php?id=88687) – they are not explicitly modeled.

DIC and TA are both explicitly modeled and are coupled to the biological growth and remineralization cycles (Moore-Maley et al., 2016). Transfer of $CO_2$ across the air-sea interface in the surface grid cell is parametrized according to Fick's second law of diffusion (Sarmiento and Gruber, 2006), using transfer coefficient (Nightingale et al., 2000), Schmidt number (Wanninkhof, 1992), and $K_0$ solubility coefficient (Weiss, 1974) parameterizations. Model pH (total scale) and $\Omega_A$ are calculated from model DIC, TA, dissolved silica, temperature, salinity, pressure, and estimated phosphate using the CO2SYS program (Lewis and Wallace, 1998) and full salinity range $K_1$ and $K_2$ constants (Millero, 2010). Phosphate is roughly approximated ($\pm 1$ $\mu$mol kg$^{-1}$, Riche (2011)) from model nitrate using the Redfield N:P ratio. Calcium ion concentrations, required for $\Omega_A$ calculation, are approximated by a linear regression to salinity (Riley and Tongudai, 1967), and by the mean observed calcium ion concentration near the Fraser River mouth (350 $\mu$mol kg$^{-1}$ (Voss et al., 2014)) where $S < 1$.

### 2.3.2 Initialization and forcing

We use profiles of temperature, salinity, fluorescence, chlorophyll $a$, nitrate, and dissolved silica (1999 through 2012) measured near the model site (Pawlowicz et al., 2007; Masson, 2006; Masson and Peña, 2009; Peña et al., 2016) (D. Masson, personal communication, 2014) to initialize the model (Moore-Maley et al., 2016). Model runs are initialized in autumn and run through a full year and then beyond until the end of the following December. Our analysis starts at the beginning of the year following the initialization date which is always longer than the 30-day spin-up period. Initial phytoplankton, zooplankton, and detritus concentrations are determined according to Moore-Maley et al. (2016). Since few DIC and TA data are available, we use a representative fall profile (11 September, 2011 (Ianson et al., 2016)) to initialize model DIC and TA. Model pH and $\Omega_A$ are not sensitive to initial carbonate chemistry conditions after the spin-up period. Time-averages (fluorescence, chlorophyll $a$, nitrate, dissolved silica) and annual fits (temperature, salinity, DIC, TA) of the initialization data near 40 m are used to set the 40 m boundary conditions (Moore-Maley et al., 2016).

The model is forced at the surface (Allen and Wolfe, 2013) by wind stress calculated from hourly wind speed and direction observed at Sandheads weatherstation (6), and by heat fluxes derived from cloud fraction, air temperature and relative humidity observed at Vancouver International Airport (5, Fig. 1b; ECCC observations, http://climate.weather.gc.ca/). Total freshwater flux (volume/time) into the Strait of Georgia is prescribed (Allen and Wolfe, 2013) using daily river discharge measurements obtained by ECCC (http://www.wateroffice.ec.gc.ca/) in the Fraser River at Hope (4) and in the Englishman River (7, Fig. 1b). Englishman River discharge is used in this study as a proxy for the contribution of small, rainfall-dominated rivers to the



freshwater budget of the Strait (Collins et al., 2009). Heat and nutrient fluxes due to freshwater are prescribed (Moore-Maley et al., 2016) as concentration × flux. Concentration values used similarly for TA and DIC are discussed in Sect. 2.4.

### 2.3.3 Evaluation

Previous studies using the model have evaluated it against physical, biological and chemical data. The vertical profiles of
density and in particular, the depth of the halocline are well represented (Collins et al., 2009). The model captures interannual variability in the biology and in the physics driving the biological variability – the model accurately predicts the timing of the spring phytoplankton bloom (Allen and Wolfe, 2013; Allen et al., 2016). The large seasonal variation of the carbon cycle is captured with some underestimation of DIC in the summer due to over-productivity (Moore-Maley et al., 2016) a common problem with coupled models in the Strait of Georgia (e.g. Peña et al. (2016)). The resulting positive-bias, root-mean-squared
error (RMSE) is 0.16 for pH and 0.51 for $\Omega_A$ (Moore-Maley et al., 2016), sufficiently small to support its use for the process studies described here.

### 2.4 Sensitivity analysis

In order to determine the sensitivity of estuarine pH and $\Omega_A$ to freshwater TA and pH, we define approximate minimum, mean, and maximum freshwater TA ($TA_f$) and pH ($pH_f$) scenarios based on the freshwater endmembers that we estimate from the
data discussed in Sect. 2.2. As seasonal variations in freshwater TA are likely flow-dependent, we define an additional three flow-dependent $TA_f$ scenarios using TA–discharge regressions of the extrapolated endmembers and the two ECCC comparison datasets discussed in Sect. 2.2. We calculate freshwater DIC ($DIC_f$) from $TA_f$ and $pH_f$ according to Sect. 2.2.

    For each $TA_f$ and $pH_f$ scenario, we ran the model for 12 separate years: 2001 to 2012 (e.g., Moore-Maley et al., 2016) to ensure a wide range of climatological forcing regimes. With 18 possible $TA_f$ and $pH_f$ combinations, we ran the model a total
of 216 times. Since the seasonality of pH and DIC in the Strait of Georgia is surface-intensified (Moore-Maley et al., 2016), we use surface (3 m average) pH, DIC:TA ratio, and aragonite undersaturation ($\Omega_A < 1$) duration as model sensitivity metrics. At constant salinity and temperature, pH and DIC:TA vary inversely. We average surface DIC:TA and pH over two different salinity regimes: $S < 20$ (summer) and $S \geq 20$ (remainder of the year). Since the model demonstrates periods of $\Omega_A < 1$ in winter and strong freshet summers (Moore-Maley et al., 2016), we evaluate $\Omega_A < 1$ duration during both seasons.

## 25  3  Results

### 3.1  Data analysis and sensitivity scenarios

The freshwater TA endmembers extrapolated from the Fraser plume cruise data (Table 1; Fig. 2) span a range of approximately 500 to 1350 $\mu$mol kg$^{-1}$ and demonstrate a significant negative correlation to discharge, although freshwater TA increases slightly above 7,000 m$^3$ s$^{-1}$ (open symbols, Fig. 3a). Observed TA from the cruise data at zero salinity ($S = 0$) are generally
slightly higher than the endmembers because of the non-conservative behavior of TA in the Fraser River at low salinity (filled


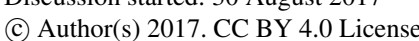


symbols, Fig. 3a). The ECCC comparison TA data from the Fraser rivermouth and ~150 km upstream at Hope described in Sect. 2.2 (Table 1) cover similar ranges and also negatively correlate with discharge (Fig. 3b and c).

Buoy pH varies seasonally from approximately 7.4 to 8.0 NIST units between low flow and peak flow (disregarding sporadic readings below 7.4 and above 8.0), and is positively correlated to river discharge except at higher flow where it reaches a

maximum (Fig. 3d). However, since the correlation is weak and the cycles of river discharge and primary productivity (low in winter, high in summer) are in-phase, the pH variability in the lower Fraser River is likely driven primarily by local processes that influence DIC such as phytoplankton growth rather than changes in river flow. The buoy pH cycle is consistent with such a biologically-driven cycle, whereas, the seasonal temperature cycle would drive pH changes in the opposite direction. The typical winter to summer temperature increase (maximum ~2 to 20°C) would cause a decrease in pH of ~0.15 pH units in

summer.

We define our three constant $TA_f$ scenarios to span the range shown by our freshwater TA endmembers: minimum $TA_f$ = 500 $\mu$mol kg$^{-1}$, mean $TA_f$ = 900 $\mu$mol kg$^{-1}$, and maximum $TA_f$ = 1350 $\mu$mol kg$^{-1}$ (Table 2). We further define our three flow-dependent $TA_f$ scenarios based on the TA–discharge, least squares regressions of the freshwater TA endmembers (Endmember Fit) and the ECCC comparison datasets (Rivermouth Fit and Hope Fit; Fig. 3a-c) for a total of six $TA_f$ scenarios

(Table 2). Our $TA_f$ scenarios cover the total observed range over multiple years, and we assume that, despite uncertainty in some of the data (Sect. 2.2), they include realistic variability.

We define our three constant $pH_f$ scenarios (Table 2) to span the range shown by the buoy pH record, however we use the oceanography-preferred (Dickson et al., 2007) total pH scale for our estuarine analysis. The present-day scenario ($pH_f$ = 7.7, total scale) is based loosely on mean NIST scale buoy pH (Fig. 3d). Given the large seasonal temperature change (>15°C

increase in summer), a constant $pH_f$ implies a summer $DIC_f$ decrease due to temperature (causing $DIC_f$:$TA_f$ to decrease by about 0.06) which is is not as large as the observed seasonal difference because observed pH increases in summer. We use a high-$pH_f$ scenario of 8.0 total scale units, which is the upper end of the observed buoy pH and is typical of present-day high TA rivers with low DIC:TA (e.g., Mississippi (Cai, 2003) and Changjiang (Zhai et al., 2007)). We suggest that this scenario represents an upper limit in the Fraser River and may represent past chemistry (lower $pCO_2$). Likewise, we define a low-$pH_f$

scenario of 7.4 units to represent possible future $pCO_2$ increases but still within the range of present-day river pH observations (buoy pH). This low-$pH_f$ scenario still has a lower $DIC_f$:$TA_f$ than low pH, weakly-buffered rivers (e.g., Kennebec (Hunt et al., 2014) and Congo (Wang et al., 2013)). Our low to high $pH_f$ scenarios (Table 2) imply $DIC_f$:$TA_f$ ratios of 1.10-1.13, 1.05-1.07 and 1.02-1.03, respectively, with the ranges being summer-winter values.

## 3.2 Sensitivity analysis

The sensitivity of model surface DIC:TA and pH to $TA_f$ and $pH_f$ is strong at low salinity ($S < 20$) which occurs during high freshwater discharge (Fig. 4b and d), and relatively weak at high salinity ($S \geq 20$; Fig. 4c and e). As the mean annual $TA_f$ increases in $TA_f$ scenarios 1 through 6 (Table 2) within a given $DIC_f$:$TA_f$ treatment (constant $pH_f$; shaded bands, Fig. 4b-g), the concurrent increase in $DIC_f$ causes model pH to decrease by raising model DIC:TA (Fig. 4b and c). This effect is strongest





at pH$_f$ 7.4 (Fig. 4b through e) where the range of model pH and DIC:TA across the TA$_f$ scenarios nearly doubles relative to pH$_f$ 8.0.

Model $\Omega_A < 1$ duration is 3 to 4 months in all winters, and about 1 month when it occurs in summer (nearly half of all runs; Fig. 4f and g). Both summer and winter $\Omega_A < 1$ duration are highly sensitive ($> 10$ day median change across all scenarios) to TA$_f$ and pH$_f$. However, despite rising estuarine DIC:TA and declining pH, $\Omega_A < 1$ duration in the estuary declines with increasing TA$_f$. The sensitivity of this decline is strongest at pH$_f = 8.0$ (where model pH sensitivity is weakest), with the largest decline between the two highest TA$_f$ scenarios (5: Hope Fit TA$_f$ and 6: Maximum TA$_f$). In the case of high pH$_f$ (8.0) in summer, the number of runs exhibiting any duration of $\Omega_A < 1$ decreased from four to one of twelve across all TA$_f$ scenarios (Fig. 4f).

The impact of flow dependence is generally weak except at pH$_f = 8.0$ where median summer $\Omega_A < 1$ duration is reduced by $\sim 10$ to 15 days (Fig. 4f) between TA$_f$ scenarios 3 and 4 (Rivermouth Fit TA$_f$ and Mean TA$_f$). This reduction is consistent with higher TA$_f$ at peak river flows in the Mean TA$_f$ case (scenario 4) compared to the Rivermouth Fit TA$_f$ case (scenario 3; Fig. 3b). Overall, modeled surface estuarine pH and $\Omega_A < 1$ duration are less sensitive to the range of pH$_f$ than they are to the range of TA$_f$. The primary importance of pH$_f$, rather, appears to be as a catalyst for pH and $\Omega_A < 1$ duration sensitivity to TA$_f$, with the sensitivity of the two quantities strongest at opposite ends of the pH$_f$ range.

## 4 Discussion

### 4.1 Two-endmember conservative mixing

To conceptualize why model estuarine pH is lowest at high TA$_f$ (while pH$_f$, or DIC$_f$:TA$_f$, is held constant), and why this sensitivity is strongest at low pH$_f$, it is useful to consider the simple case of conservative mixing between freshwater and seawater endmembers. In the absence of biological processes and gas exchange, physical dilution alone drives linear mixing between these two endmembers, and thus the mixing curves for DIC, TA, temperature, dissolved silica, and phosphate are linear with respect to salinity (e.g., DIC and TA, Fig. 5a). However, pH calculated from these curves (CO2SYS (Lewis and Wallace, 1998)) with full salinity range $K_1$ and $K_2$ constants (Millero, 2010)) is not linear with salinity and demonstrates a characteristic minimum between the two endmembers when pH$_f$ is lower than the seawater endmember pH (Fig. 5b). Similar theoretical salinity-pH curves have been calculated (Mook and Koene, 1975; de Mora, 1983; Whitfield and Turner, 1986; Hofmann et al., 2009; Hu and Cai, 2013) and also observed (e.g., Scheldt River estuary (Mook and Koene, 1975; Hofmann et al., 2009), Fraser estuary (de Mora, 1983)).

Estuarine pH along the mixing line is clearly sensitive to pH$_f$, however large differences also arise from changes in TA$_f$ (Fig. 5b). The reason for this sensitivity to TA$_f$ is the dramatic difference in DIC:TA along the mixing line between TA$_f$ cases despite an equal ratio between the two cases at the freshwater and seawater endmembers (Fig. 5c). In the high TA$_f$ case, the estuarine DIC:TA decreases more linearly with salinity resulting in a higher estuarine DIC:TA and lower estuarine pH overall. In the low TA$_f$ case, estuarine DIC:TA decreases rapidly with salinity producing a higher estuarine pH. The difference between

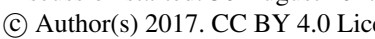



these curves increases with the difference between the freshwater and seawater DIC:TA endmembers (increasing $DIC_f$:$TA_f$, decreasing $pH_f$).

Despite our limited model results in the low salinity range ($S < 10$), the simple mixing case can help us explain the model sensitivity. In both the model case and the mixing case, higher $TA_f$ (at constant $DIC_f$:$TA_f$) produces lower estuarine pH (Fig. 4d and e and 5b) because of the dramatic increase in estuarine DIC:TA (Fig. 4b and c and 5c). The low $pH_f$ (high $DIC_f$:$TA_f$) scenarios demonstrate the strongest estuarine pH sensitivity to $TA_f$ because they produce the largest difference between freshwater and seawater DIC:TA endmembers (Fig. 5c).

Ocean pH and $\Omega_A$ are often assumed to be coupled, however they are clearly not in brackish estuarine waters fed by rivers with moderate or low $TA_f$ (Fig. 4d through g). This asynchrony arises because the response of estuarine carbonate ion over the large range of river $TA_f$ and $pH_f$ scenarios is more sensitive to changes in total DIC than shifts in the equilibrium point of the carbonate system. Thus increases in model DIC (and carbonate ion) shorten $\Omega_A < 1$ duration despite reductions in estuarine pH. This carbonate mechanism is also responsible for the increased sensitivity of model $\Omega_A$ to $TA_f$ at high $pH_f$. At low $pH_f$, the carbonate equilibrium shifts are strong enough to stabilize $\Omega_A < 1$ duration to $TA_f$ ($DIC_f$) changes, but at high $pH_f$ these equilibrium shifts are weak and $DIC_f$ ($CO_3^{2-}$) increases dominate.

Our model sensitivity studies demonstrate the impact of freshwater carbonate chemistry on the marine environment beyond the simple bulk mixing case. Air-sea gas exchange, biological activity, dynamic mixing, and estuarine circulation make these studies more realistic, yet also more complicated. Hofmann et al. (2009) found that adding a biogeochemical model to a prescribed physical mixing scenario in the highly polluted Scheldt estuary decreased the low-salinity pH minimum due to strong remineralization of abundant anthropogenic organic carbon (Frankignoulle et al., 1996) and increased pH at higher salinity due to intense outgassing (Schiettecatte et al., 2006). In contrast the Fraser River estuary is biologically productive and deep (model site is deeper than 300 m) and much of the organic matter produced locally sinks and is remineralized well below the surface layer (Johannessen et al., 2008) resulting in increased estuarine pH across the salinity range during summer (Ianson et al., 2016) compared with the two-endmember mixing model. Still, this simplified case reproduces the overall estuarine pH sensitivity that we observe in the coupled model.

## 4.2  Implications for other estuaries

Present-day carbonate chemistry in world rivers covers a large range (DIC or TA from ~100 to 7000 $\mu$mol kg$^{-1}$; Fig. 6). Pollution drives part of this variability (black symbols, Fig. 6) as does latitude, in large part from the presence of carbonate rocks in many temperate drainage basins (Amiotte Suchet et al., 2003). Seasonal biological productivity may also play a strong role in some temperate rivers by drawing down DIC, at least near the river mouth. In the Fraser River the productive season occurs during high flow keeping DIC:TA low (pH high), while other rivers experience an increase in DIC:TA from low flow to high flow (Fig. 6c, d). The low flow DIC:TA ratio in the Fraser is thus uniquely high (Fig. 6c), although it is less able to exert influence on neighboring ocean waters due to a smaller river plume and storm-induced winter mixing. DIC and TA in most rivers are strongly flow (or seasonally) dependent, particularly the tropical rivers (Fig. 6).





The sensitivity of estuarine pH to this flow dependence (seasonal $TA_f$ variability) in the simple mixing model changes significantly with freshwater DIC and TA ($\Delta$pH, Fig. 5d). This sensitivity is large ($\Delta$pH > 0.1) and centered at low salinity (8-10) over the Fraser River $TA_f$ range (500-1,350 $\mu$eq kg$^{-1}$, yellow line), but is greatly reduced ($\Delta$pH < 0.05) and shifted to higher salinity (15-20) over a higher (but similar width) $TA_f$ range approximately representative of the Mississippi River (2115-2870 $\mu$eq kg$^{-1}$, solid black line; Table 3). Likewise, $\Delta$pH is high ($\Delta$pH > 0.1) and at low $S$ (4-5) for a lower $TA_f$ range representative of the tropical Amazon River (246-549 $\mu$eq kg$^{-1}$, teal line; Table 3), and is reduced ($\Delta$pH < 0.04) over a similar $TA_f$ range to the Amazon but shifted slightly higher to represent the Arctic Lena River (651-860 $\mu$eq kg$^{-1}$, black dash line; Table 3).

For our endmember mixing curves (Fig. 5) across an even larger range of mean $TA_f$ (0-6000 $\mu$eq kg$^{-1}$), overall estuarine pH sensitivity ($\Delta$pH) to a given $TA_f$ range ($\Delta TA_f$) decreases (Fig. 7a) and salinity at maximum $\Delta$pH increases (Fig. 7b) with increasing mean $TA_f$ ($\overline{TA_f}$). While these dependencies are estimated using endmembers specific to the Fraser-Strait of Georgia system, we propose that these relationships have broader implications for rivers and their neighboring estuaries. Specifically, mean river TA ($\overline{TA_f}$) primarily determines estuarine pH sensitivity ($\Delta$pH and $S$ at maximum $\Delta$pH) to river TA variability ($\Delta TA_f$). As such, low TA rivers like the Kennebec (yellow triangle right), Ob (white circle), and Yenisey (white square, Fig. 7) may have larger seasonal $\Delta$pH variations at lower salinities than high TA rivers like the Changjiang (black circle) and Mackenzie (white diamond), despite sharing similar $\Delta TA_f$. Conversely, rivers with similar $\Delta TA_f$:$\overline{TA_f}$ ratios (e.g., Lena (white triangle up), Mackenzie (white diamond), Changiang (black circle), Mississippi (black diamond); Fig. 7a) may share similar $\Delta$pH maxima as the increase in $\overline{TA_f}$ offsets the increase in $\Delta TA_f$, but salinity at maximum $\Delta$pH would still increase with $\overline{TA_f}$ independently of $\Delta TA_f$ (Fig. 7b).

Thus, for arbitrary rivers sharing the same Strait of Georgia seawater endmember, the estuaries of high TA rivers experience smaller seasonal estuarine pH ranges than those of low TA rivers over similar seasonal freshwater TA range widths. However, these high TA river estuaries also experience their zones of strongest seasonal pH sensitivity at higher salinities than their low TA counterparts, which likely impact a larger areal extent of the estuary (this zone is different than the zone of maximum pH sensitivity to changes in DIC alone which occurs at DIC$\sim$TA (Egleston et al., 2010)). In this regard, seasonal pH changes in the Strait of Georgia driven by mixing alone are large ($\Delta$pH > 0.15) because the Fraser (red star, Fig. 7a) is relatively low in TA but has a relatively high seasonal TA range relative to other world rivers. However, this large pH range is centered below salinity 10 so the impact on the estuary may not be as severe.

### 4.3 Implications for future climate

In the coming decades, weathering rates and mean river TA are unlikely to change significantly due to climate (Riebe et al., 2001). In contrast, increasing atmospheric $p$CO$_2$ will increase DIC in the ocean and most likely in rivers as well. River DIC$_f$ may be influenced by many additional variables, including changes in freshwater flow, human pressures (local anthropogenic inputs) and anticipated increases in river temperature. For example, the present day glacial DIC and TA end-member yields a high DIC:TA ratio relative to most world rivers (Fig. 6c, d), but is in equilibrium with the atmosphere at a river temperature of 0°C (Meire et al., 2015). If this pure glacier water were to experience a 10°C increase during its passage to the ocean, then




outgassing would decrease $DIC_f$ by about 10% ($\sim$9 $\mu$mol kg$^{-1}$) if it remained in atmospheric equilibrium. However, its pH would stay about the same (slight increase) in this scenario.

While the present study finds a maximum estuarine pH sensitivity to river TA at relatively low salinity ($\sim$10 for the Fraser–Strait of Georgia endmembers, Fig. 5c), DIC changes to the system independent of TA generally impact estuarine pH most severely where DIC $\sim$ TA (Egleston et al. (2010), $S \sim$ 12-20 for the Fraser–Strait of Georgia), creating a salinity zone of particularly strong sensitivity (Hu and Cai, 2013). These future DIC increases will shift this zone to higher salinities. As the surface areas of various salinity zones in an estuary generally increase away from the river, this shift would increase the areal size of this highly sensitive zone at present-day river flows.

A warming climate has (Zhang et al., 2001) and will continue (Morrison et al., 2014) to reduce the peak freshet flows of glacial, temperate rivers and move the freshet timing earlier. Winter flows may increase, which may increase the sensitivity of estuaries to river chemistry during these low flow times beyond our model (currently insensitive) predictions. However, the surface Strait of Georgia is relatively acidic compared with oceanic extrapolations of fresh water endmembers from the otherwise similar, glacially-fed temperate Corcovado estuary in Chile (Torres et al., 2011), despite the Fraser having higher alkalinity than what the Puelo River appears to have. In fact the surface Strait of Georgia is already aragonite-undersaturated for the whole winter season (Fig. 4g). Unsurprisingly, the river has the strongest impact during years with the highest river flows. These years have the lowest summer pH and $\Omega_A$ when river carbonate chemistry remains constant (not shown, (Moore-Maley et al., 2016)). Thus as climate change continues to reduce peak flows, the impact of rivers on neighboring estuaries may decrease overall despite changing river carbonate chemistry.

## 5  Conclusions

The Fraser River is moderately alkaline (TA = 500–1350 $\mu$mol kg$^{-1}$) due to chemical weathering, however, its DIC:TA ratio is high during winter (low flow), even relative to rivers that are considered heavily polluted. While it contributes less freshwater and TA to the Northeast Pacific ocean than does the Columbia River, it exerts strong influence over a large, semi-enclosed estuarine region that is vulnerable to climate change and DIC-rich relative to the open ocean. We assessed recent and historic observations of the carbonate system in the river and its estuary, the southern Strait of Georgia. We then investigated the sensitivity of surface estuarine pH and $\Omega_A$ to river TA and pH (18 scenarios) using a predictive biogeochemical model over more than a decade of present-day physical forcing scenarios with strong interannual variability.

Estimated freshwater TA endmembers extrapolated from Strait of Georgia cruise data and Fraser River TA observations near the mouth decrease with increasing river flow, as is the case in most temperate rivers. However, pH is highest during peak flow despite this low alkalinity and significant summer warming, likely due to biological uptake of DIC. This seasonal variation means that, unlike some other rivers (e.g., Congo), the Fraser River is most acidic (high DIC:TA) during low flow when river impact on the estuary is at a minimum. The Fraser is always well undersaturated with respect to aragonite ($\Omega_A \leq 0.1$), typical of the more acidic Arctic and tropical rivers.





Within the Fraser's Strait of Georgia estuary, including our model site (~25 km seaward of the river mouth), the river plays a key role in driving strong seasonal cycles in physical and biogeochemical variables. For roughly three months during summer, surface salinity is low and variable, ranging from 5–20, depending on strength and timing of the Fraser freshet. In contrast, winter conditions in the estuary are less variable and more saline, typically near 25 (always $< 30$). Like in the Fraser, surface

pH in the Strait of Georgia is high during the summer freshet (8.1–8.35) when biological productivity throughout the Strait is also high, as is typical of temperate systems. This summer estuarine pH is sensitive to both river TA and DIC:TA ratio (pH), but large DIC fluxes associated with the summer productivity reduces the sensitivity to the latter. Winter estuarine pH is always around 8.0 and relatively insensitive to river TA and DIC:TA ratio since river flow is at a minimum.

During freshets when salinity is low ($< 20$) in neighboring estuaries, surface $\Omega_A$ becomes decoupled from pH due to the

impact of calcium and carbonate ion dilution. Thus, despite high productivity in the southern Strait of Georgia (high pH) during summer, there are long periods (up to 40 days) of surface $\Omega_A$ undersaturation during strong freshets, which occur in nearly half of the years studied. These periods occur regardless of river carbonate chemistry, excepting the single highest freshwater TA and pH scenario, which represents only the strongly alkaline rivers at present day like the Mississippi, Changjiang, and Mackenzie. The duration of these undersaturated periods is less sensitive to river TA at high river DIC:TA (low pH) than is

estuarine pH. However, low river DIC:TA (high pH) allows river TA to exert more control, decreasing this duration and even its incidence, as may have been the case in many estuaries, including the Fraser, in the past.

While the sensitivity of estuarine pH to changes in river TA is low for alkaline (high TA) rivers, the salinity zone of this sensitivity is high and potentially covers large areal extents. Conversely, the greater sensitivity of estuarine pH for low TA rivers is maximized at low salinities and thus may be confined to smaller sections of the estuary. However, in the future as

atmospheric $CO_2$ continues to dissolve into aquatic systems, estuarine pH sensitivity to river TA will increase in the estuaries of moderately alkaline rivers like the Fraser because of rising DIC:TA. These DIC:TA increases may also reduce the ability of rivers like the Fraser to buffer against estuarine surface aragonite undersaturation. Additionally, like the zones of maximum estuarine pH sensitivity to freshwater TA, the zones of maximum estuarine pH sensitivity to DIC changes (DIC~TA) will be pushed to higher salinities as well, likely affecting larger areal extents of estuaries, depending on changes in river flow.

*Code and data availability.* Model source code and run scripts will be hosted by the UBC Salish Sea bitbucket repository at https://bitbucket. org/salishsea prior to publication. Results files for the sensitivity experiments will be hosted at the Abacus Dataverse Network which is a research data repository of the British Columbia Research Libraries' Data Services http://dvn.library.ubc.ca/dvn, prior to publication. Model initialization data is available from the Fisheries and Oceans Canada Institute of Ocean Sciences Data Archive http://www.pac.dfo-mpo.gc. ca/science/oceans/data-donnees/search-recherche/profiles-eng.asp. Carbon cruise data from the Fraser River plume presented in Ianson et al.

(2016) will be hosted at the U.S. Department of Energy's ESS-DIVE data repository prior to publication. Carbon cruise data from the Fraser River plume presented in de Mora (1983) are published entirely in de Mora (1981). All other data presented in this article are available from their cited URL locations.



*Competing interests.* The authors declare that they have no conflict of interest.

*Acknowledgements.* This work was supported by a Natural Sciences and Engineering Research Council of Canada (NSERC) Discovery grant to the third author, Fisheries and Oceans Canada's Climate Change Science Initiative and International Governance Strategy programs, and the Marine Environmental Observation, Prediction, and Response (MEOPAR) Network of Centres of Excellence of Canada. We thank

5  Sophia Johannessen and Robie Macdonald at the Fisheries and Oceans Canada Institute of Ocean Sciences (IOS) for sharing their Fraser Plume carbon data, Diane Masson and Peter Chandler (IOS) for sharing their Strait of Georgia survey data, and Eleanor Simpson and Karen Kohfeld (Simon Fraser University), Marty Davelaar (IOS), Yves Perrault (Little Wing Oysters) and Andre Comeau (Okeover Organic Oysters) for collecting and sharing data from the Freke Ancho River. We thank Doug Latornell at UBC for developing the model working environment, and Robie Macdonald and Paul Covert for providing helpful insights. The second author thanks Niki Gruber for his hospitality

10  at ETH-Zurich while this manuscript was completed. We also acknowledge three anonymous reviewers for their detailed feedback.





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





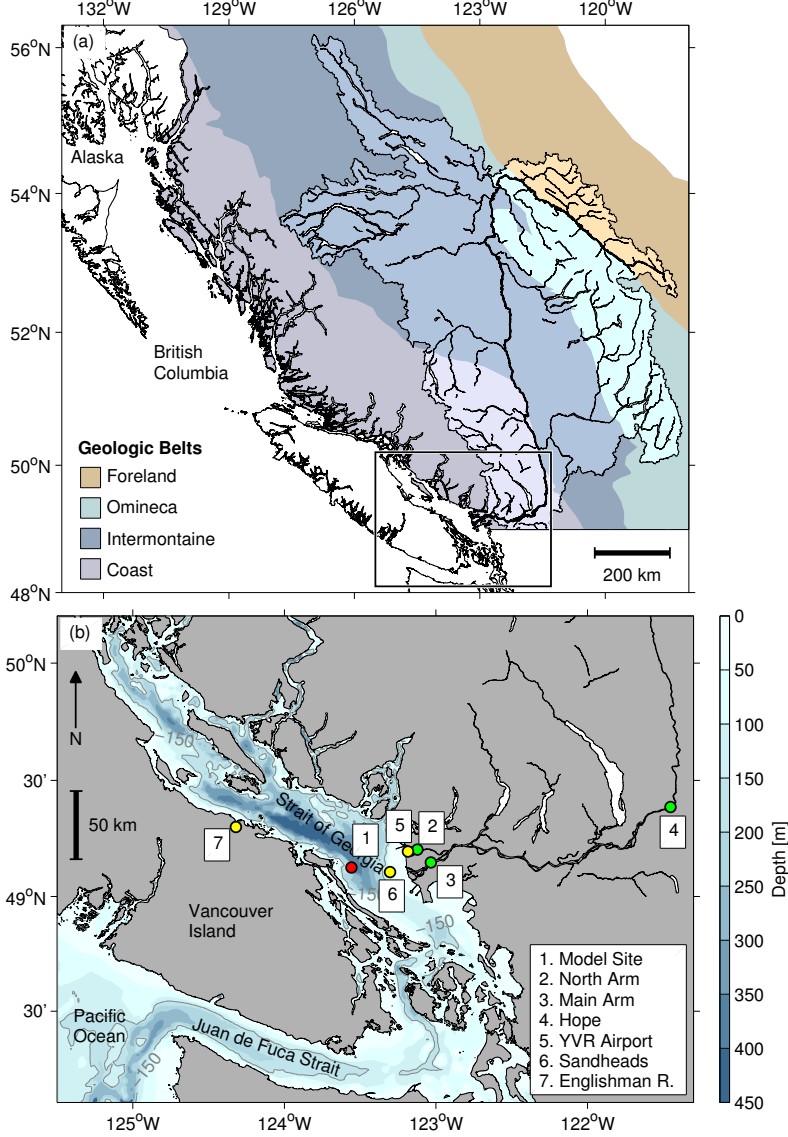

**Figure 1.** Maps of (a) the Fraser River watershed with geologic belts (Wheeler et al., 1991), and (b) the Fraser River delta and Strait of Georgia with Environment and Climate Change Canada (ECCC) carbon chemistry sampling sites (2-4) in green, ECCC meteorological stations (5-6) and river gauging stations (4, 7) in yellow, and the model location (1) in red. TA measurements (2-4) and pH readings at the ECCC Fraser River Water Quality Buoy (3) are used to constrain Fraser River carbon chemistry (Table 1). The model is forced with hourly windspeed (6) and meteorlogical (5) observations (http://climate.weather.gc.ca/), and daily Fraser (4) and Englishman (7) River discharge (http://wateroffice.ec.gc.ca).





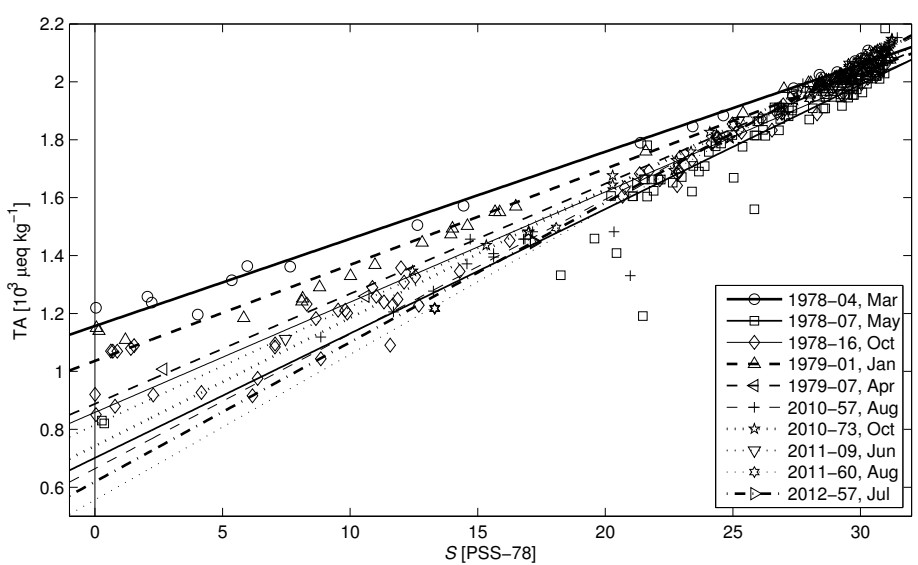

**Figure 2.** TA versus salinity ($S$) regressions from several cruises near the Fraser River plume prior to 1980 (de Mora, 1983) and post-2009 (Ianson et al., 2016). Cruise ID numbers (legend) begin with the sampling year. Each cruise contains at least one datapoint at $S < 20$. TA is generally conservative with $S$ except near $S = 0$ (de Mora, 1983) – observations where $S < 0.1$ are excluded from the regressions. Extrapolated TA values to $S = 0$ are used as freshwater TA endmember estimates (Table 1).




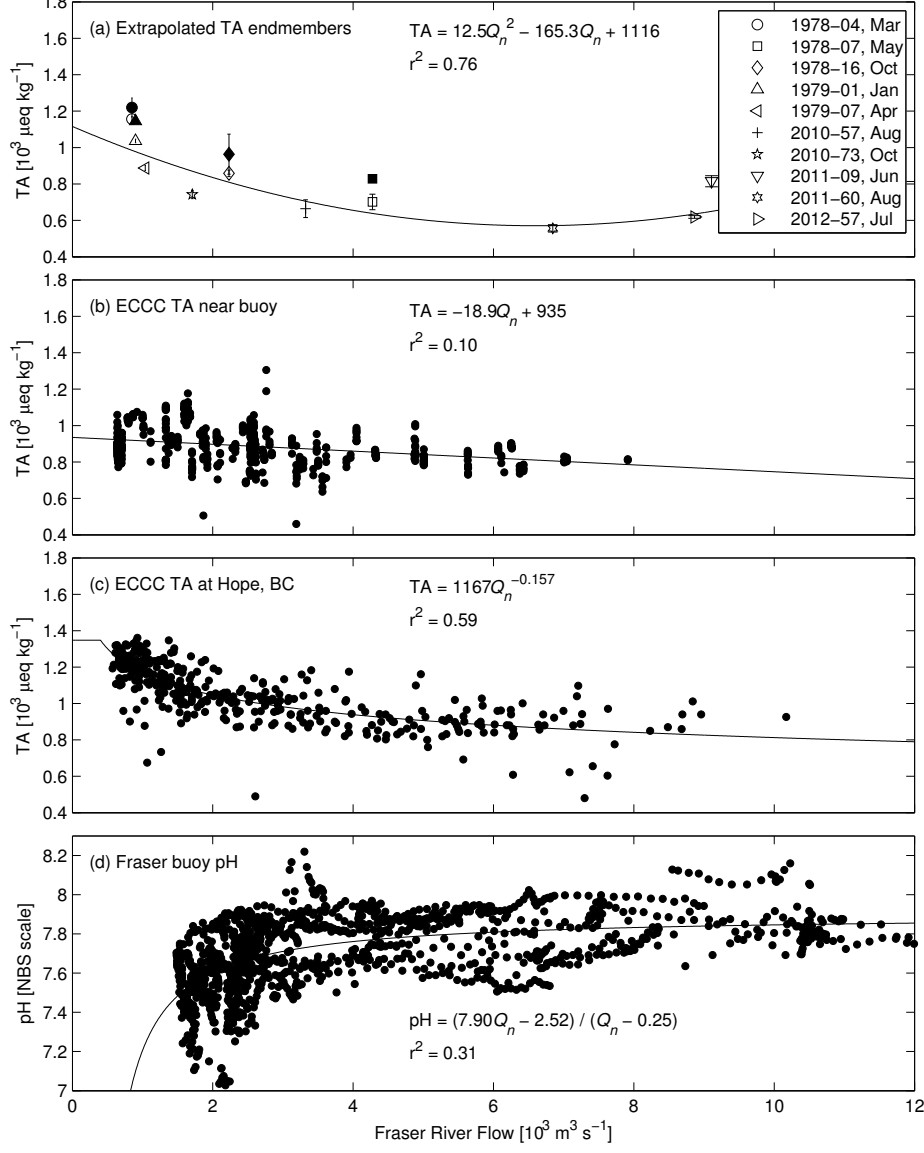

**Figure 3.** Fraser River TA (a) extrapolated endmembers, (b) observations (ECCC) near the river mouth, and (c) observations (ECCC) at Hope, BC, and (d) ECCC buoy pH observations (Table 1) all versus Fraser River discharge near the mouth estimated (Pawlowicz et al., 2007) from flow observations at the Hope ECCC flow gauge (http://www.wateroffice.ec.gc.ca/; Fig. 1a). Solid symbols (a) are the observed TA at $S = 0$, where available, from the corresponding open symbol cruise extrapolations. Cruise numbers (a) are the same as in Fig. 2 and correspond to de Mora (1983) prior to 1980 and Ianson et al. (2016) post 2010. The fits shown are least-squares regressions where $Q_n = Q/1000$ m$^3$ s$^{-1}$, $Q$ being the estimated discharge near the river mouth. The regressions are used to define flow-dependent TA scenarios for testing model pH and $\Omega_A$ sensitivity.





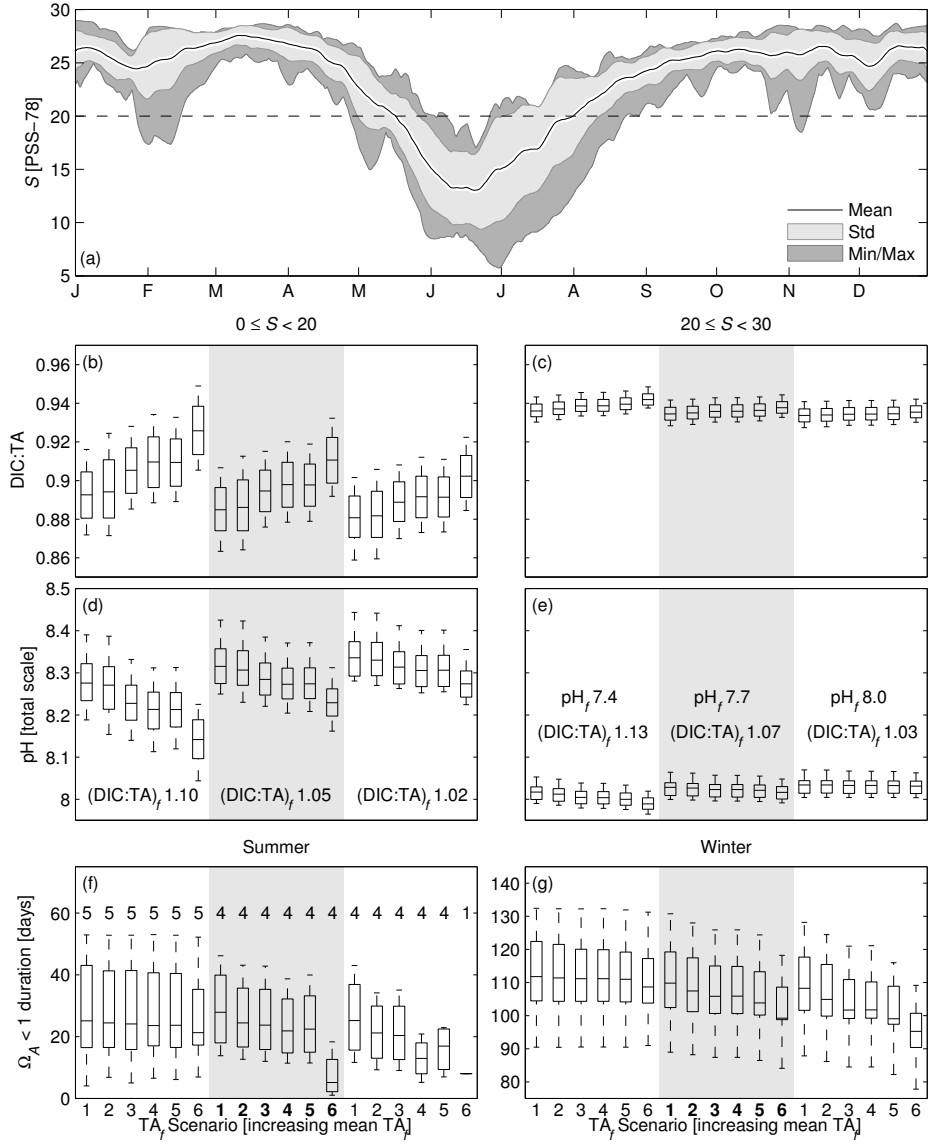

**Figure 4.** Model (a) mean, standard deviation, and min/max $S$ calculated across the period 2001-2012, and 2001-2012 boxplots of estuarine (b, c) DIC:TA and (d, e) pH averaged over (b, d) $S < 20$ and (c, e) $S \geq 20$, and (f) summer and (g) winter estuarine $\Omega_A < 1$ duration with 6 increasing mean annual $\text{TA}_f$ scenarios (horizontal axis, Table 2) for each of 3 $\text{pH}_f$ scenarios (grey/white sections) moving left to right from low $\text{pH}_f$ (high $\text{DIC}_f$:$\text{TA}_f$) to high $\text{pH}_f$ (low $\text{DIC}_f$:$\text{TA}_f$). Model $S$, DIC, TA, pH and $\Omega_a$ are 0-3 m averages. Each box indicates the median, standard deviation about the mean, and 95% confidence intervals across the 12 years of runs. In (f), each box only contains runs that exhibited $\Omega_A < 1$ in summer (total labelled above each box).



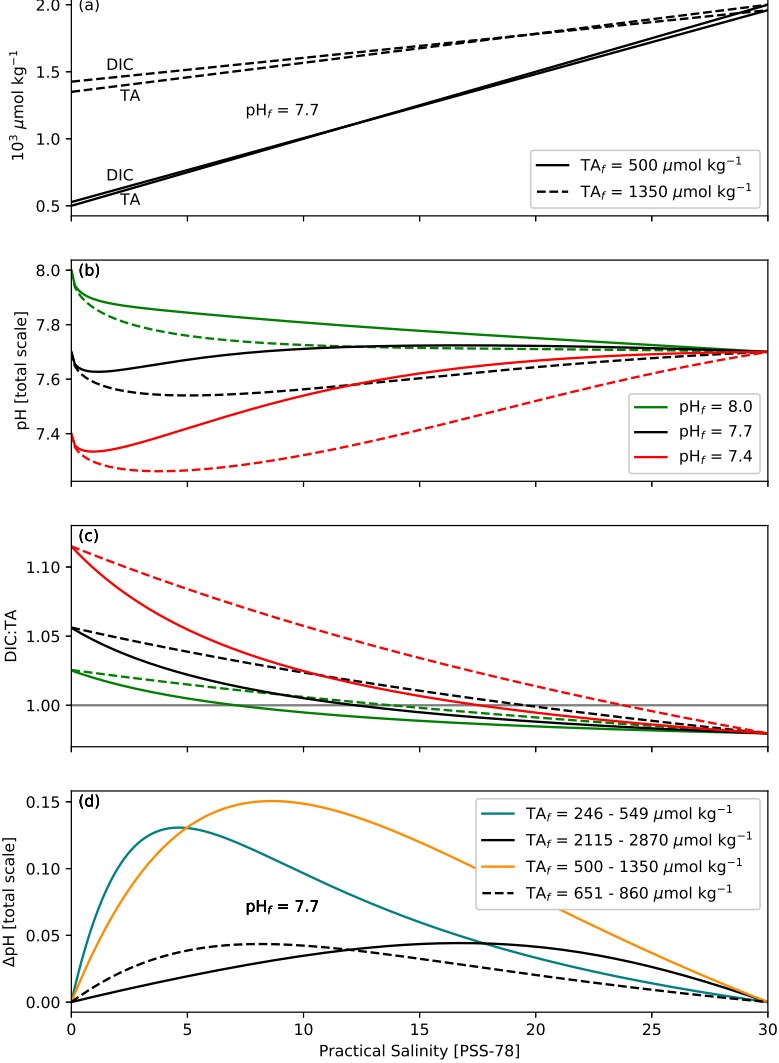

**Figure 5.** Theoretical two-endmember mixing curves of (a) DIC and TA (for $pH_f$ = 7.7 only), (b) pH, and (c) DIC:TA for high flow ($TA_f$ = 500 $\mu$eq kg$^{-1}$, solid line) and low flow ($TA_f$ = 1350 $\mu$eq kg$^{-1}$, dashed line) scenarios at $pH_f$ of 7.4 (red), 7.7 (black), and 8.0 (green). (d) Seasonal estuarine pH differences ($\Delta$pH) between high flow and low flow $TA_f$ scenarios (low and high $TA_f$, respectively; $pH_f$ = 7.7) for four theoretical $TA_f$ ranges (Table 3): a tropical range similar to the Amazon (teal), a polluted range similar to the Mississippi (black), a temperate range similar to the Fraser (yellow), and an Arctic range similar to the Lena (black dash). All mixing curves (a-d) share the same Strait of Georgia seawater properties defined as the mean 40 m model DIC, TA, T, S, Si, and PO$_4$ values (Moore-Maley et al., 2016), and the same freshwater $T$, Si, and PO$_4$ values defined as the mean observations near the Fraser River mouth between 2009 and 2011 (Voss et al., 2014). DIC$_f$ and pH are calculated using CO2SYS (Lewis and Wallace, 1998) and full $S$-range $K_1$ and $K_2$ dissociation constants (Millero, 2010).



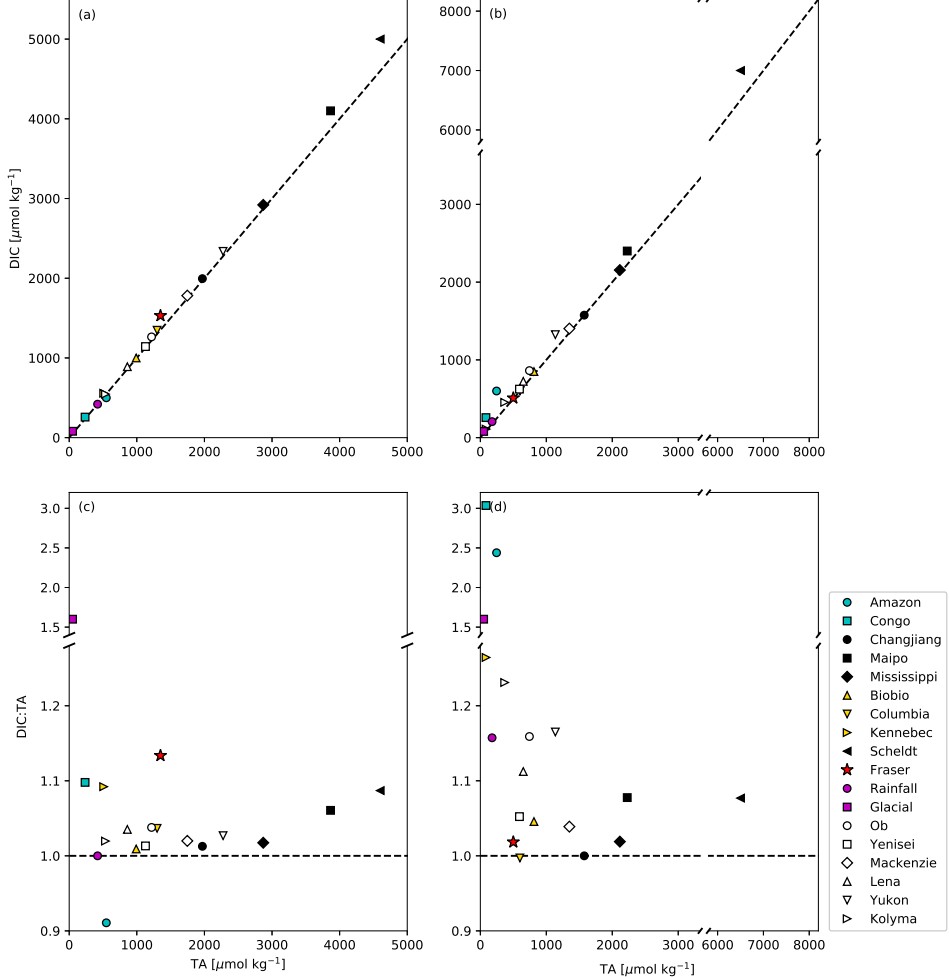

**Figure 6.** DIC and TA literature values for selected world rivers (Table 3) at (a, c) low river flow (generally high TA) and (b, d) high river flow (generally low TA). The dash line is the DIC:TA 1:1 line. The legend is in order of increasing latitude. Cyan symbols represent tropical watersheds, black symbols represent urbanized/polluted watersheds, yellow symbols represent temperate watersheds, magenta symbols represent watershed type proxies, and white symbols represent Arctic watersheds. The Fraser River is emphasized as a red star. DIC ranges are calculated first from $p\text{CO}_2$, next from pH when DIC observations could not be found (Table 3). TA ranges are calculated similarly when unavailable (Maipo and Biobio rivers only). For most rivers, DIC:TA can be significantly greater than 1 (c, d) despite falling visually close to the 1:1 line (a, b). Uncertainty is not shown and not always reported.





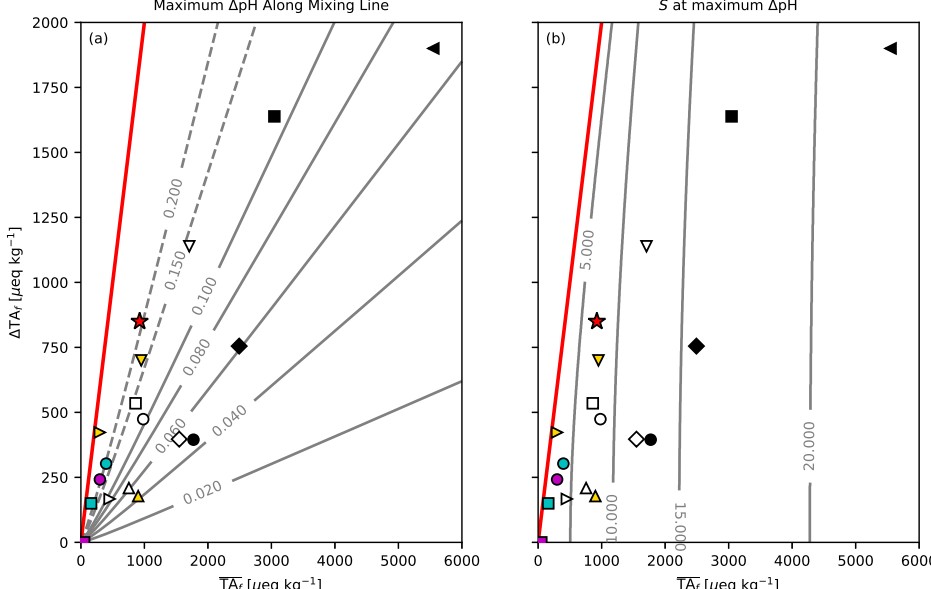

**Figure 7.** Contours (light gray) of the magnitude (a) and $S$ (b) of maximum seasonal estuarine pH differences ($\Delta$pH) between high-low flow, two-endmember mixing curves (Fig. 5d) at $\mathrm{pH}_f$ = 7.7 as a function of mean freshwater TA ($\overline{\mathrm{TA}_f}$) and freshwater TA range ($\Delta\mathrm{TA}_f$). Red lines in (a) and (b) indicate the region where $\overline{\mathrm{TA}_f} - \Delta\mathrm{TA}_f/2 < 0$ (i.e., negative lower $\Delta\mathrm{TA}_f$ limit). For reference, $\overline{\mathrm{TA}_f}$–$\Delta\mathrm{TA}_f$ pairs for the Amazon (teal circle), Congo (teal square), Changjiang (black circle), Maipo (black square), Mississippi (black diamond), Biobio (yellow triangle up), Columbia (yellow triangle down), Kennebec (yellow triangle right), Scheldt (black triangle left), Fraser (red star), Rainfall Proxy (magenta circle), Glacial Proxy (magenta square), Ob (white circle), Yenisey (white square), Mackenzie (white diamond), Lena (white triangle up), Yukon (white triangle down), and Kolyma (right triangle right) Rivers (Table 3) in order of increasing latitude are also shown (a and b). All mixing curves share the same Strait of Georgia seawater endmember defined as the mean 40 m model DIC, TA, T, S, Si, and PO$_4$ values (Moore-Maley et al., 2016), and the same freshwater T, Si, and PO$_4$ defined as the mean observations near the Fraser River mouth between 2009 and 2011 (Voss et al., 2014). DIC$_f$ and pH are calculated using CO2SYS (Lewis and Wallace, 1998) and full $S$-range $K_1$ and $K_2$ dissociation constants (Millero, 2010).



**Table 1.** Datasets used to assess the range and variability of freshwater TA and pH near the Fraser River estuary.

| Dataset | Locations(s)[a] | Years | Frequency | Qty | Units[b] | Method |
|---|---|---|---|---|---|---|
| Endmember TA[c] | See de Mora (1983) | 1978 – 1979 | seasonal | TA | mmol L$^{-1}$ | Anderson and Robinson (1946) |
| | See Ianson et al. (2016) | 2010 – 2012 | seasonal | TA | $\mu$mol kg$^{-1}$ | Dickson et al. (2007) |
| Buoy pH[d] | Main Arm | 2008 – 2013 | hourly | pH | NIST units[e] | Ethier and Bedard (2007) |
| Hope TA[f] | Hope | 1979 – 1999 | semi-monthly | TA | mg CaCO$_3$ L$^{-1}$ | ECCC (2006) |
| Rivermouth TA[f] | North and Main Arm | 2004 – 2009 | semi-monthly | TA | mg CaCO$_3$ L$^{-1}$ | ECCC (2006) |

[a] See Fig. 1.

[b] TA converted to $\mu$mol kg$^{-1}$ for use in the present study (Sect. 2.2).

[c] de Mora (1983); Ianson et al. (2016).

[d] ECCC Fraser River Water Quality Buoy (http://aquatic.pyr.ec.gc.ca/RealTimeBuoys/fraserRiverChart.aspx).

[e] National Institute of Standards and Technology scale.

[f] Environment and Climate Change Canada (ECCC, http://aquatic.pyr.ec.gc.ca/webdataonlinenational).





**Table 2.** SoG TA$_f$ scenarios based on fits between Fraser River and SoG TA data and the estimated (Pawlowicz et al., 2007) Fraser River discharge, $Q$, near the delta (Fig. 3).

| | Scenario | TA$_f$ ($\mu$eq kg$^{-1}$)[a] | $Q$ limit[b] |
|---|---|---|---|
| 1 | Minimum TA$_f$ | 500 | |
| 2 | Endmember Fit | $1116 - 165.3Q_n + 12.5Q_n^2$ | $< 11{,}000$ |
| 3 | Rivermouth Fit | $935 - 18.9Q_n$ | |
| 4 | Mean TA$_f$ | 900 | |
| 5 | Hope Fit | $1167Q_n^{-0.157}$ | $> 400$ |
| 6 | Maximum TA$_f$ | 1350 | |

[a]$Q_n = Q/1000$ m$^3$ s$^{-1}$

[b]Designates the range of $Q$ (m$^3$ s$^{-1}$) where the scenario is valid (closest valid point used beyond this range).





**Table 3.** Ranges (low flow to high flow) of TA, DIC, pH, $pCO_2$, and $T$ in several world rivers, in order of increasing latitude.

| River | TA ($\mu$eq kg$^{-1}$) | DIC ($\mu$mol kg$^{-1}$) | pH (NBS) | $pCO_2$ ($\mu$atm) | $T$ (°C) | Location (km upstream) | Frequency | Time range | Source |
|---|---|---|---|---|---|---|---|---|---|
| Amazon | 549–246[a] | – | – | – | – | delta | monthly | 1963–1964 | Gibbs (1972) |
| Congo | 235–85 | 500–600[b] | 6.5–7.2 | – | – | 400 | 8 cruises | 1982–1984 | Richey et al. (1990) |
| Changjiang | 1970–1575 | 258 | 6.7–5.7 | 2018–6853 | 28 | 350 | monthly | 2011 | Wang et al. (2013) |
| | – | 1995–1575 | 8.0–7.8 | 607–1395 | 7–29 | 60 | 4 cruises | 2003–2006 | Zhai et al. (2007) |
| Maipo | – | 4100–2400 | 7.4–7.5 | 6700–3500 | 23–12 | – | – | – | – |
| Mississippi | 2870–2115 | 2920–2155 | – | – | – | delta | 2 cruises | Aug–Sep 1998 | Cai (2003) |
| | – | – | – | – | 8–30 | 100 | daily | 1983–2012 | White and Visser (2016) |
| Biobio | – | 1000–850 | 8.1–7.7 | 550–700 | 29–9 | – | – | – | – |
| Columbia | 1300–600 | – | – | – | 3–24 | 100 | monthly | 1995–2012 | Evans et al. (2013)[c] |
| | – | – | – | 735–176 | – | estuary | 8 cruises | 2007–2008 | Evans et al. (2013) |
| Kennebec | 510–87 | 557–110 | 4.9–7.0 | 1771–203 | 27–0 | 0–50 | monthly | 2004–2008 | (Hunt et al., 2014) |
| Scheldt | 4600–6500[b] | 5000–7000[b] | 7.9–7.5 | 4000–14000 | 25–3 | 81–96 | monthly | 1996–1999 | (Hellings et al, 2001) |
| Fraser | 1300–500 | – | 7.4–8.0 | – | 4–20 | Table 1 | Table 1 | Table 1 | Table 1 |
| Ob | 1218–744[a] | – | 8.0–7.2 | – | –2–17 | – | regularly[e] | 2009–2011 | Arctic-GRO I[d] |
| Yenisey | 1128–593[a] | – | 8.3–7.7 | – | 1–13 | – | regularly[e] | 2009–2011 | Arctic-GRO I[d] |
| Mackenzie | 1747–1350[a] | – | 8.2–7.8 | – | 0–15 | – | regularly[e] | 2009–2011 | Arctic-GRO I[d] |
| Lena | 860–651[a] | – | 8.0–7.4 | – | 0–11 | – | regularly[e] | 2009–2011 | Arctic-GRO I[d] |
| Yukon | 2276–1137[a] | – | 8.1–7.2 | – | 0–15 | – | regularly[e] | 2009–2011 | Arctic-GRO I[d] |
| Kolyma | 536–369[a] | – | 8.2–7.1 | – | 0–10 | – | regularly[e] | 2009–2011 | Arctic-GRO I[d] |

[a] Reported as HCO$_3^-$ in mg L$^{-1}$. Converted using 60.0168 g HCO$_3^-$ mol$^{-1}$ molecular weight and approximating density as 10$^3$ kg m$^{-3}$.

[b] Reported as $\mu$mol L$^{-1}$ or $\mu$eq L$^{-1}$. Converted by approximating density as 10$^3$ kg m$^{-3}$.

[c] United States Geological Survey (USGS) National Stream Quality Accounting Network (http://water.usgs.gov/nasqan/).

[d] Arctic Great Rivers Observatory I constituent data (NSF-1107774, http://www.arcticgreatrivers.org/data.html).

[e] 15 comprehensive campaigns with a focus on freshet, late summer, and under-ice periods; daily samples over the freshet.