# Peer review of "The sensitivity of estuarine aragonite saturation state and pH to the carbonate chemistry of a freshet-dominated river"

_Biogeosciences, 2017_

## Referee Comment (RC1) · Anonymous Referee #1 · 10 Nov 2017

This paper deals with the effect of variable boundary conditions in a river on the estuarine pH and saturation state. It does so by applying a previously described model that is used to run a large number of scenarios of feasible riverine conditions. Obviously the subject is an important one, and the tools used, modeling, are suitable to achieve the goals in the manuscript. However, I found this paper particularly difficult to read and to keep focus on the findings that it describes. In the end I even wonder what it is that I have learned here that I did not already know.

The reasons for this are diverse: First of all, too much information is being compressed in this manuscript, and in some of the figures - e.g. figure 4 is particularly difficult to

interpret. A few well-chosen scenarios would have been much easier to explain and to depict. Secondly, too little information is given about the system under study, so that it is not clear what processes actually might actually produce the patterns or how relevant these findings are for other systems. Thirdly, the model description is too vague - it is even unclear if the 1-D model resolves the vertical extent (which I think it does) or has the dimension arranged along the estuarine length axis (which I think it should).

The 2008 paper from Salisbury et al, that is used to back up the scarcity of papers on estuarine carbonate chemistry is outdated by 10 years, and there are indeed some recent papers on this subject that are not mentioned in the manuscript, e g. Volta et al., 2015 (Hydrology and Earth System Sciences), Cai et al, 2017 (Nature Communications) to name a few. There is also older work e.g. Regnier et al., 1997- marine Chemistry, that deals with (modeling of) pH in estuaries and that are not mentioned in the manuscript.

Finally, while the paper shows that, under some conditions of freshwater influence, the estuarine pH and saturation appears more sensitive, it is not clear why this is so. Of course, carbonate chemistry is a difficult discipline, but procedures to formalize the attribution of processes on pH shifts have been developed in the past and were recently put in a consistent framework by Hagens and Middelburg, 2016, Geochem. Comochim. Acta 187. The absence of a discussion that untangles the importance of processes on the modeled pH changes in a more quantitative way makes it hard to grasp the relevance of these results for other systems.

So in conclusion, while the subject is an important one, the way the manuscript is structured does not lead to a large enough increase in insight for this paper to be accepted in its current state.
* * *

---

## Referee Comment (RC2) · Anonymous Referee #2 · 30 Nov 2017

I must admit that I am conflicted in making a recommendation of this paper. First, this is my favorite subject and I like the approach of a combination of data and modeling (also a combination of numerical model and simple mixing model). However I do not think the combination is successful. My dissatisfaction is further aggravated by the frustration in reading it at every step. First, the paper doesn't present much new data. I believe most new data and the numerical model were published in their two earlier publications (authors really need to say what is new here). Second, the coupled physical and biological ROMS model (first part of the paper) and the simple mixing (second part) are not very compatible. I like that they want to extrapolate the study to a global discussion. However their "transport" of the Fraser conditions to other rivers are clearly

not a good idea. What I like least is the assumption of all rivers have a greater than 1.02 DIC:TA ratio. Many of the problems of this paper came from this assumption but the authors actually generalized their only partially correct conclusions. For example, the first sentence in the beginning of the Discussion (p.8, line 18-20) says: "To conceptualize why model estuarine pH is lowest at high TAf, . . .." First, this statement is only true in the situation the authors created that is the ratio of river DIC:TA = 1.02 to 1.1. In this case, the higher the TAf, the more DIC is in the river water, so river water pH is low. Also it will take more seawater (that is at a higher salinity) to reduce the DIC:TA ratio to a certain value. Thus it is not "high TAf" but a high (DIC-TA) or DIC:TA in river water that is important here and is the reason behind the phenomenon. In the low TAf case, the same DICf:TAf ratio only means a very small DIC in excess of TA in the river water, and it can be very quickly eliminated during river-ocean mixing, or in other word, the pH of the estuarine water is dominated by the seawater. I refer the authors to the paper by Liang et al. 2017 (Marine Chemistry). Further, in extrapolation of the results, the authors didn't consider temperature effect but this effect can also be significant in controlling carbonate system speciation.

Second, the conditions derived from the Fraser River were based on highly uncertainty pHf and TAf data. As the authors correctly recognized that in some rivers organic contribution to the TA can be serious. If the examples cited by the authors are also true in the Fraser River (e.g., as high as 90% of TA is organic alk), then, how can we believe the DICf calculated from the pHf and TAf? Even if Org-Alk is 25%, it would be a serious problem. In the current paper, the extrapolation of the Fraser River DIC:TA ratio globally is just not appropriate. I am not necessarily against making overly simplified assumptions (sometimes one has to), but please fully assess the uncertainty of your assumptions. Here the assumption of DICf:TAf > 1.02 probably not just changes the result slightly it perhaps will change the major conclusion derived. Regarding the data, the authors must clearly say what is new that are not published in their two earlier papers.

[Figure]

Third, the paper is poorly prepared and hard to follow (see my detailed reading notes). Fig. 4 is particularly hard to understand or guess. The authors also often write sentences that are seemingly correct but actually are not so or not clear; this would confuse the readers. For example, p.9, line 10, it says "This asynchrony arises because the response of estuarine carbonate ion over the large range of river TAf and pHf scenarios is more sensitive to changes in total DIC than shifts in the equilibrium point of the carbonate system." First, I don't understand what this sentence really says. Second, it sounds like to suggest that one can change all three parameters (TAf, pHf and DICf) at the same time. If here "DIC" is not river DIC but internal estuarine, biologically modified DIC, this is probably true, but the authors didn't say that. There are many places, the writing is not transparent to me. So a thorough rewrite with a better readability is also needed.

In the abstract (p.1, line 10), it says "rivers with high DIC and TA produce lower estuarine pH due to an increased estuarine DIC:TA ratio, but higher estuarine OmegaA because of DIC contributions to the carbonate ion." I like this statement. But I do not really see the result description and an extensive discussion of in the text body. Does this indicate that authors have changed mind a bit on exactly what they want to focus on in this this paper?

I do not like the implications for future climate. You have speculated too much! (To say high atmospheric CO2 will increase river pCO2 is simply wrong as river pCO2 is generally so much higher than the atm-pCO2.) So simplify it and merge it in the Discussion with just a few sentences.

Finally your summary is too long and repeats too much of the Discussion.

If the authors can address these serious issues reasonably, I feel this paper can be a good contribution to our field. Here are my suggestions (not sure I really need to do this): If the numerical model study of the Fraser case is new (say it), perhaps expand the model description and limit your discussion to this case, which is essentially what

you did but just don't called global extrapolation.

In addition to the above summary, I also wrote down some notes while reading the text twice (some places many times). Some may be trivial or not fully correct.

p.2, Line 15-16, Is this true? I have not seen a river whose TA is NOT flow-dependent. Perhaps, it is because the West paper is about the silicate weathering. Not sure if this is also true for carbonate weathering.

p.4, line 15-18, Carbonate Alk is about 50% in the Satilla River Georgia (Cai et al. 1998). Again Cai et al. (1998) paper should be cited as it is the first study of this issue.

p.4, Line 15-18, this really worries me. If TA data quality is so bad and the organic contribution is so large, how do we know the rest of modeling is correct?

p.6, Is RMSE = 0.16 pH unit a small uncertainty? It appears quite large to me. Perhaps you need to put it in the context of overall pH change.

p.7, line 11 says "We define our three constant TAf scenarios", then line 17 says "We define our three constant pHf scenarios (Table 2)" (first there is no pHf in Table 2). I am quite confused if these are related or separated assumptions? From my comment below on Fig. 5 caption, I don't think you need to call them "constant TAf or pHf", just river endmember scenario 1 to 6 is enough. They have nothing to do with whether TAf is constant with river discharge; they are just your scenarios.

p.7, line 20, I do not understand this– "Given the large seasonal temperature change (>15_C increase in summer), a constant pHf implies a summer DICf decrease due to temperature (causing DICf :TAf to decrease by about 0.06)"—why increased temperature leads to DICf decrease? Is this just a decreased solubility effect or increased river pCO2 leads to more CO2 degassing? Do you mean that under constant pHf and Alkf, higher temperature leads to lower DICf? That is true. But the real question is what is really controlling the pHf at constant TAf but allowing DICf to decrease? You can't just set a certain parameter constant arbitrarily. Anyway you need to explain your as-
sumption. Reading Fig. 5 caption (b), I finally see how you did it. You selected two TAf values one low and one medium-high. Then at each TAf, you selected three pHf values and calculated 6 DICf values. Then you take these 6 TAf and DICf combinations and mix them with the same seawater (TAsw and DIC sw) to generate the 6 mixing lines. I assume at least b and c should be based on these 6 same simulation. (For panel d, it seems you used the variable TAf.) My question is– are these combinations realistic? I now take your 6 combinations and put them into CO2SYS (I assume T=15C and all other acid = 0) and verified/confirmed your DICf and DICf:TAf. No problem except that all DICf:TAf ratio > 1.02. This doesn't make sense to me. Perhaps it is true for the low TAf rivers (but in these river the org-ALK is often large), DIC:TA < definitely occurs in medium and high TA rivers.

p.7, line 30-33, It is very hard to understand what the authors try to say after reading this part and Fig. 4 many times. Are DIC:TA (Fig. 4b) or pH (Fig. 4d) averaged over the estuary or what? Very frustrating.

p.8, the description of Fig. 4 is not very clear. Figure 4 is not clearly labeled. Some supplement instructions are needed. (Like does all gray and white sections in figure 4b, 4c, 4d mean different pH scenario? What does the "5 and 4" mean in figure 4f)

p. 8, I am shocked that line 16 moves into Discussion. Now I realized that the entire ms is essentially a model study of pH sensitivity to river discharge and (DIC:TA)f ratio (the latter is also a function of discharge). Then, I went back to read their two earlier papers. The mode and the field data were already presented there. The authors need to say what is new of the first part of this paper comparing with the earlier papers.

p.8, line 18-20, see main point.

p.8, line 28 to p.9, line 8, this is true. In the high river TA estuarine, DIC:TA is decreased slow whereas in low river TA estuarine water, DIC:TA is quickly modified and dominated by the seawater ratio. If this is the only point this paper wants to talk then why presenting the numerical model? Some of the other recent papers also talked about this point.

However the assumption of same DIC:TA ratio for high TAf and low TAf rivers are likely problematic. Not sure how meaningful is this scenario simulation.

Page 9, Line 8: "Ocean pH and $\Omega A$ are often assumed to be coupled": Some references may be needed here to support this. I think the word "coupled" and "decoupled" are misused here. In some sense, pH and omega are always coupled. They are just not "coupled" in a simple way as our "intuitions" may suggest. The simple case here is that [Ca2+] increased as salinity increases, but that has no direct effect on pH. There are other more subtle factors or processes influencing pH and omega differently. However I do not think we can simply call that as "decoupled". You can find a better name.

p.9, line 12-14 is not clear.

p. 10, I like the discussion on seasonality, but it is based on physics (TAf and discharge only).

p.10, line 14-16, true that high river TAf systems like the Mississippi provide a strong buffer effect and its delta-pH shouldn't change as much as that in low TAf systems during mixing. However biological production could raise pH to a very high value in the Mississippi.

Fig. 6 c &d, Note there are two arms of DIC:TA ratio to TA with a minimum at seawater TA. There is nothing magic here but the authors should mention the reason. The left reflects the mixing between the generally high ratio in river with a low ratio in seawater. The right arm reflects the mixing of a few very high TA rivers (with TA higher than the seawater) with seawater.

p.11, line 1 and 6 are not consistent. Warming allow more CO2 degassing and decreases river DIC. Yes, I agree. But increased atmospheric CO2 probably won't increase DIC as river DIC is so much higher than the atm-pCO2. If there is any increase it likely increases TA equally (through increase of weathering rate).

p.10-11, I think this section is very speculative and should be deleted or combined into

earlier discussion with short sentences. These speculations do not help, e.g., we do not know which competing factor will dominate and if river DIC will increase. Overall as the authors agree that these effects are rather small comparing with eutrophication induced surface biological production and subsurface respiration induced pH changes.

p.11-12, I generally do not like long summary, which essentially does more repeating discussion.

Finally, regarding pH scale. I do not understand why the authors switches between NBS and total scales. I'll stick with one and note there is big uncertainty in either one when salinity is extreme (that is pHT doesn't work for river water and pHNBS doesn't work fully for seawater). Also, pH was given as in "NIST units." This is not the right way we marine chemists will say. It should be in "NIST scale" (I would just call it "in NBS scale"). When saying a pH change then you can say "a change of 0.xx pH units". There is not such a name called NIST or NBS or total units. It is scale!

Since the ms is an open access discussion paper, I also asked a colleague who knows statistics better than me to read it. Below is her comments. I have read these and generally agreed.

Page 6 Line 10: the positive bias, root-mean squared error of the model output is 0.16 for pH and 0.51 for the saturation state of aragonite. The authors claim that these errors are sufficiently small to support the model use for the process studied in the paper. I am not sure about this claim. The error of 0.51 looks big enough to me from my understanding of acidification impacts. The model may be good in reproducing the physical field and biology bloom as the authors stated here.

A regression with R2 of 0.1 without a P value is impossible for readers to judge whether the regression is significant or not (Fig. 3b). If the regression is not significant, TA is not flow dependent, and then the model can't use this relation to derive a scenario. For the regression-based scenarios, since data vary greatly, uncertainties associated with these regressions should be provided and transferred to the model outputs. Without

knowing the uncertainties, considering the error of the model output of the saturation state of aragonite, 0.51, not so small, it is hard to evaluate the duration of ïĄÜA <1 in any scenario. The authors may specify how low ïĄÜA is in these scenarios.

References Xue, L., Cai, W. J., Sutton, A. J., & Sabine, C. (2017). Sea surface aragonite saturation state variations and control mechanisms at the Gray's Reef time-series site off Georgia, USA (2006-2007). Marine Chemistry. https://doi.org/10.1016/j.marchem.2017.05.009.

---

## Editor Comment (EC1) · J. Middelburg (Editor) · 1 Dec 2017

Dear Dr. Moore-Maley:

Thank you for submitting your paper to Biogeosciences (Discussion).

As you have seen two referees have handed in their report and both expressed founded and balanced reservations on the quality and readability of your paper. The bottom line is that your paper (1) lacks focus: too many items are addressed and essential information is often lacking, (2) does not incorporate essential recent key papers: i.e. your results need to better put in context, and (3) does not articulate clearly what is new

and what has been published before. The referees provided many useful suggestions to improve and focus your paper.

I propose that you provide a detailed rebuttal to the comments of the referees and additionally upload (as a rebut to this short editors' feedback) an outline of a revised paper that resolves the issues identified. On the basis of that outline I will decide whether this manuscript will be further considered for Biogeosciences or not. If further considered it will go through at least one other full round of external evaluation.

With best regards, Jack Middelburg, Associate Editor

―――――――――――――――

---

## Author Comment (AC3) · 24 Dec 2017

**Editor Response**

**Summary**

We appreciate the detailed feedback provided by yourself and both reviewers and would enjoy the opportunity to revise the manuscript in accordance. We agree that we have extended some of our analyses beyond their scope and in doing so have unwittingly lost focus in the manuscript. We propose that we scale back to focus our analyses only on the Fraser River. We will still place it in the context of other global rivers (Figure 6) however we will remove the end-member analysis that relied on DIC and TA from other world rivers (Figures 5-d and 7).

Specifically, we will provide additional detail concerning the Fraser and its estuary, including what is known and less known regarding the drivers of the inorganic carbon cycle. The river is a key driver and yet we have few reliable carbon data in the fresh and brackish waters in the study region. This paucity provides a strong motivation for our analysis. By clarifying this motivation in the text, the key results will be highlighted. We will include the additional references (not all of which were available at the time that this manuscript was submitted) that the reviewers have suggested, where appropriate. We also will define new sensitivity scenarios to include more recent and newly acquired data where possible and reduce (and sometimes remove) the dependence on data in which we have less confidence (such as data collected using outdated methods or river TA with high organic alkalinity uncertainty). We will re-run all the simulations with these new scenarios and produce a new sensitivity summary figure (4), with clarified presentation, to reflect these new results. Finally, we will strengthen the delivery of our main findings and highlight the importance of using the biogeochemical coupled model by refocusing our results (3), discussion (4), and conclusion (5) sections to target the key points below.

**Key points**

1. Responses of estuarine pH and $\Omega_A$ to Fraser River DIC-TA are asynchronous and strongest at opposite ends of the Fraser DIC:TA range.

2. Seasonal estuarine productivity reduces estuarine pH sensitivity to river chemistry during summer

3. Future Fraser River flow regimes with lower flow in the biologically productive season will favor lower estuarine pH and $\Omega_A$, but the river will dominate a smaller areal region in the estuary.

**List of proposed revisions**

I Tighten writing and improve clarity throughout

II Add more study area background

III Reference recent and reviewer-recommended literature

IV Add more details of the model configuration including

- a motivation for the vertical domain rather than a horizontal domain along the estuary axis

- how the vertical formulation accounts for estuarine circulation
- how the simulations presented here are different from previously published results

V Add most recent TA and pH observations and remove all older TA data collected using outdated/uncertain methods or freshwater TA measurements where organic alkalinity uncertainty is high

VI Use only the total scale for pH - convert all freshwater pH to total scale for presentation in the manuscript

VII Revise freshwater endmember scenarios based on recently available new estuarine data, re-run sensitivity experiments, and remake sensitivity summary figure (4)

VIII Discuss river endmember chemistry in terms of DIC-TA rather than TA, and DIC:TA rather than pH

IX Add a timeseries figure of several (selected) individual years of model salinity, pH and $\Omega_A$ and overlay the result(s) of these single year runs on the whisker objects in the summary figure (4). This addition will clarify how figure (4) summarizes the > 200 model sensitivity runs, 12 *for each* river chemistry scenario.

X Add selected model DIC, TA, DIC:TA, and pH at corresponding salinities on the theoretical end-member mixing plots in figure (5) so that the effect of sources and sinks in the estuary can be seen relative to pure mixing only.

XI Remove world river endmember analysis and figures 5d and 7

XII Discuss our results in the context of buffer factors (e.g., Hagens and Middelburg 2016, Hu and Cai 2013, Elgeston et al. 2010), and clarify why we do not use them

**Proposed outline**

(proposed additions, )

**1 Introduction**

Additional references (not all of which were available at the time that this manuscript was submitted) that the reviewers have suggested, will be cited here or where appropriate (**REVISION III**)

- **Motivation to study carbonate chemistry in estuaries**
  - Importance of carbonate chemistry in estuaries. We will acknowledge the new work by Cai et al. 2017, Xue et al. 2017 etc.
  - Complexity of estuarine systems and carbon chemistry within them
  - More explicit introduction of buffer factors and future sensitivity (add Hagens and Middleburg 2016)

- **River DIC and TA drivers**

- Seasonal river flow variability and its effect on carbonate chemistry
- Relationship to weathering with global context
- Influence of pollution/anthropogenic nutrients (add Cai et al. 2017)

- **Study premise**

  - Identify study region and lay out sensitivity analysis
  - Clarify that the model is vertical, parametrizes estuarine circulation, and contains phytoplankton/zooplankton functional groups (**REVISION IV**)
  - Revise description of data used (**REVISION V**)
  - Clarify paucity of accurate data as a motivation for the study

**2  Methods**

**2.1  Study Area (REVISION II)**

- **Figures**

  - Map (Fig 1 old numbering)
  - Hydrograph of Fraser (NEW FIGURE)

- **Describe the Fraser River and Strait of Georgia**

  - Seasonality, interannual variability, productivity, carbonate chemistry (Collins et al. 2009, Allen and Wolfe 2013, Moore-Maley et al. 2016, Ianson et al. 2016, Pawlowicz et al, 2007.)
  - Decoupling of productivity and respiration (deep fjordic system - respiration signal not seen at surface)

- **Fraser River carbonate chemistry**

- **Organic alkalinity discussion** (moved from 2.2)

**2.2  Data**

- **Figures**

  - TA vs salinity with extrapolation to S=0 (Fig 2 old numbering, + 2014, 2016, 2017 data **REVISION V**)
  - TA and pH vs river discharge (Fig 3 old numbering, , + 2014, 2016, 2017 data **REVISION V**)

- **Describe TA datasets used**

  - Ianson et al. 2016 (+ 2014, 2016, 2017 data previously unpublished)
  -  (uses outdated methods)
  -  (uses outdated methods and likely contains significant organic alkalinity)

- **Describe extrapolated endmembers**

- **Describe method for determining river DIC:TA**

    - Buoy pH (present in total scale **REVISION VI**)

    - Discuss pH sensor accuracy and provide errorbars for DIC:TA uncertainty

    - Discuss temperature dependence of DIC:TA calculation

-  (moved to 2.1)

**2.3 Model**

**2.3.1 Overview**

The physical model configuration is described and the biological and carbon model detail are given. In the new version we will also describe and justify the vertical 1-D model, provide more details of the physical part of the model including estuarine circulation and the importance of vertical mixing (**REVISION IV**)

**2.3.2 Initialization and forcing**

A description of how the model is initialized, the bottom boundary conditions and the surface forcing used to drive the model are given.

**2.3.3 Evaluation**

We describe how the model was evaluated and the statistics of that evaluation. In the new version we will discuss the implications of the evaluation. In particular, we will clarify that although RMSE is not small, the positive bias prevents overestimating severity of corrosive conditions

**2.4 Sensitivity analysis**

- **Describe the freshwater carbonate chemistry scenarios.**

    - We will define two new sets of scenarios: one independent of river flow and one dependent on river flow. Although it is intuitive to vary river DIC and TA across the scenarios, the impact is due to DIC-TA and DIC:TA. Thus for each set of scenarios we will provide a table with all four of DIC, TA, DIC-TA and DIC:TA for each scenario. (**REVISION VIII**)

    - Instead of basing our flow dependent scenarios on statistical fits to our observations, we will construct these flow dependent scenarios based on what we expect the flow dependent curves to look like across the range of observations summarized in section 2.2. (**REVISION VII**)

- **Describe the model runs**

    - Each scenario will be run across all of our years, giving more than 200 total simulations.

– We will provide more detail about the different annual forcing combinations instead of referencing Moore-Maley et al. 2016 (which used a single river chemistry scenario) to emphasize uniqueness of this study.

**3 Results**

**3.1**

(Moved to sections 2.2, 2.4, 3.2)

**3.2 Sensitivity analysis**

- **Figures (REVISION IX)**

    – NEW FIGURE: timeseries of selected individual years (from $> 200$ different runs) of each of model forcing, model salinity, pH and $\Omega_A$ to show model behavior at low and high river flow, in extremes in river chemistry

    – Boxplot (Fig 4, old numbering): remake figure to show new sensitivity runs, add years shown in the new timeseries figure overlaid in the box and whisker objects

- **Describe the seasonal and interannual dynamics of salinity, pH, and $\Omega_A$ according to the new model timeseries figure (REVISION IX)**

- **Introduce the reader to Figure 4 (old numbering) including what the boxes mean and how the model scenarios are organized along the axes (REVISION VII)**

- **Highlight the key points in the results:**

    – Trends of the boxes across all scenarios (**KEY POINT 1**)
    – Length of each box across the 12 different years (**KEY POINT 2**)

**4 Discussion**

**4.1 Two-endmember conservative mixing (Fraser only)**

- **Figures**

    – Salinity space plots  (Fig 5, old numbering) (**REVISION XI**)

    – Add results from summary figure (Fig 4, old numbering) at corresponding salinities in Fig 5 (old numbering) (**REVISION X**)

- Discuss similarities of model results and mixing exercise We will make this text more concise and target **KEY POINT 1**

- Separate importance of biogeochemical sources and sinks within the estuary from simple mixing scenarios (add reference to Regnier et al. 1997) **KEY POINT 2**

- Relevance of DIC/TA ratios (add reference to Xue et al. 2017) (**REVISION III**)

- Compare/contrast our endmember mixing example with the use of buffer factors (e.g. Hagens and Middleburg 2016) and briefly explain why we don't use them (**REVISION XII**)

**4.2  (REVISION XI)**

-

  -
  -

-

**4.3 Implications for future climate (REVISIONS I and III)**

- Merge this section with Fraser comparison to world rivers

  - Place DIC/TA world rivers (Fig 6, old numbering) here

- Discuss predictions for future flow regimes We will make this discussion more quantitative by referencing the new timeseries figure and Fig 4 (old numbering) from the results section (3.2) **KEY POINT 3**

- Discuss sensitivity to increasing T (add Hagens and Middleburg 2016)

- Discuss impact of future increases in atmospheric $CO_2$ scenarios on estuaries (Volta et al. 2016)

**5 Conclusions (REVISION I)**

We will rewrite the conclusions in truncated form to (1) restate our study premise from the final paragraph of the introduction, and (2) highlight our key points

---

## Author Response (AR1)

**Editor Response**

We greatly appreciate the opportunity to submit a revised manuscript. The reviewers shared many serious concerns about the content and presentation of our first submission, and we have made our strongest effort to address those concerns as thoroughly as possible. To that end, we have completely reworked this manuscript at every stage. We have isolated our data analysis to our own TA dataset from the Strait of Georgia, including new, previously unpublished profiles from our ongoing sampling program, and the Fraser River Water Quality Buoy pH record maintained by Environment and Climate Change Canada. Using only these data, we have produced a new suite of scenarios and executed new model runs to account for these changes. We have produced entirely new figures that clarify our model runs and our key points. Finally, we have rewritten the majority of our manuscript, particularly the results, discussion, and conclusions, to more clearly articulate our findings and emphasize our key points. We no longer attempt to extrapolate our results to other estuaries and instead only place our results in global contexts that are already well-established from existing literature. We feel that the revision process has strengthened this manuscript immensely, and we are grateful to the reviewers for their time and detailed comments.

**Reviewer 1 Response**

(*Reviewer Comments*, Response, Proposed Changes, **Added Changes**)

1. *This paper deals with the effect of variable boundary conditions in a river on the estuarine pH and saturation state. It does so by applying a previously described model that is used to run a large number of scenarios of feasible riverine conditions. Obviously the subject is an important one, and the tools used, modeling, are suitable to achieve the goals in the manuscript. However, I found this paper particularly difficult to read and to keep focus on the findings that it describes. In the end I even wonder what it is that I have learned here that I did not already know … while the subject is an important one, the way the manuscript is structured does not lead to a large enough increase in insight for this paper to be accepted in its current state.*

   We appreciate the detailed feedback provided by yourself, the other reviewer and the editor, and would enjoy the opportunity to revise the manuscript in accordance. We agree that we have extended some of our analyses beyond their scope and in doing so have unwittingly lost focus in the manuscript. We propose that we scale back to focus our analyses only on the Fraser River. We will still place it in the context of other global rivers (Figure 6) however we will remove the end-member analysis that relied on DIC and TA from other world rivers (Figures 5-d and 7).

   Specifically, we will provide additional detail concerning the Fraser and its estuary, including what is known and less known regarding the drivers of the inorganic carbon cycle. The river is a key driver and yet we have few reliable carbon data in the fresh and brackish waters in the study region. This paucity provides a strong motivation for our analysis. By clarifying this motivation in the text, the key results will be highlighted. We will include the additional references (not all of which were available at the time that this manuscript was submitted) that the reviewers have suggested, where appropriate. We also will define new sensitivity scenarios to include more recent and newly acquired data where possible and reduce (and sometimes remove) the dependence on data in which we have less confidence (such as data collected using outdated methods or river TA with high organic alkalinity uncertainty). We will re-run all the simulations with these new scenarios and produce a new sensitivity summary figure (4), with clarified presentation, to reflect these new results. Finally, we will strengthen the delivery of our main findings and highlight the importance of using the biogeochemical coupled model by refocusing our results (3), discussion (4), and conclusion (5) sections to target the key points below.

   **Key points**

   (a) Responses of estuarine pH and $\Omega_A$ to Fraser River DIC-TA are asynchronous and strongest at opposite ends of the Fraser DIC:TA range.

   (b) Seasonal estuarine productivity reduces estuarine pH sensitivity to river chemistry during summer

   (c) Future Fraser River flow regimes with lower flow in the biologically productive season will favor lower estuarine pH and $\Omega_A$, but the river will dominate a smaller areal region in the estuary.

- We have removed the endmember discussion for other world estuaries beginning on page 14, line 15 and ending on page 15 line 8.
- We have added additional background about the study area starting on page 3: line 32
- We have addressed additional references in the introduction on page 2: lines 9, 11, 16, 17, 20, 28, and 29, and in the discussion on page 16: lines 5 and 33
- We have completely reworked the analysis beginning with new data and analysis (Figs. 2 and S1, Table 2), new model runs and analysis (Figs. 3–7) and strengthened presentation throughout.
- We have clarified our main findings and made them consistent throughout the manuscript in abstract (page 1, lines: 11-19), introduction (page 3: lines 11-14), results (page 11: lines 17-25), discussion (page 12: lines 30-35, page 13: lines 1-12, page 16, lines: 24-34) and conclusions (lines 17-32)

2. *... figure 4 is particularly difficult to interpret. A few well-chosen scenarios would have been much easier to explain and to depict.*

Figure 4 depicts our key results (which will be clarified/focused) from the model sensitivity study, and is critical to the paper. We have constructed new scenarios based on current (rather than old/suspect) data. We plan to improve the accessibility of this figure by strengthening its description in the text (Section 3.2), and by making the following modifications

- The new scenarios will simply be numbered, with a table of associated freshwater pH, TA, DIC:TA, and DIC-TA values. We will plot the box and whisker objects against these scenario numbers in order of increasing freshwater TA or DIC:TA
- We will add a figure that shows a few selected years as timeseries plots and explain how these timeseries map to points on Figure 4 to clarify where this synopsis figure comes from.
- We will overplot selected individual years of salinity (a), DIC:TA (b, c), pH (d, e), and aragonite undersaturation duration (f, g) on top of the salinity climatology (a) and box statistics (b-g). These individual years will illustrate how each year fits into the box plot.

- We have created a new summary figure (Fig. 6) based on new scenarios and model runs. We now plot each model run individually and clearly distinguish it from the other runs using color, and exploring two selected runs (2010 and 2012) for additional scrutiny.
- We have added 3 new figures leading up to our main summary figure (Fig. 6) including a timeseries figure (Fig. 4) of selected model runs to illustrate the temporal behavior of the model before proceeding to the salinity averages. The runs from the timeseries plots are clearly indicated on Fig. 6 (stars)

- **We have moved the salinity panel to its own plot with the river hydrograph also shown (Fig. 3). Again we have plotted the individual years as traces instead of the envelope around the mean curve.**
- **We have completely reworked our discussion of these plots, taking the time to carefully explain each one while remaining clear and concise**

3. *... too little information is given about the system under study, so that it is not clear what processes might actually produce the patterns or how relevant these findings are for other systems.*

   We wrote our study area section (2.1) with the intention of introducing the relevant processes in the Fraser-Strait of Georgia system for later discussion, but we now agree with the reviewers that the level of background presented and the degree to which that background is addressed in the discussion are inadequate. Specifically, it is important for the reader to recognize that the Fraser is globally significant (largest Pacific-draining river in Canada) and strongly seasonal, yet confined to a long residence time in the estuarine Strait of Georgia by tides and topography, the results of which are strong seasonal stratification, productivity, and near-surface aragonite undersaturation in the Strait. These processes are all resolved or parameterized in the model and present fundamental differences between the modelling results in Section 3.2 and the endmember mixing exercise in Section 4.1. Furthermore, these processes are not equally important in all estuaries and thus provide indicators for the applicability of this study to other systems. In the new manuscript, we will refocus Section 2.1 so that it supports the narrative that we have described here.

   **We have added additional information about the study area to highlight the importance of the river and biology in determining the existing carbonate chemistry of the system starting at Page 3: line 33.**

4. *it is even unclear if the 1-D model resolves the vertical extent (which I think it does) or has the dimension arranged along the estuarine length axis (which I think it should).*

   The model is 1-D vertical. This vertical model was used rather than an estuarine length axis model because local phytoplankton seasonality is more sensitive to the wind and light climatology than to the river (but the river is still important). The vertical mixing model resolves these wind (stratification within a deep fjordic system) and light effects on phytoplankton mechanistically. In contrast, a (1-D) horizontal model would have to parameterize these effects. We will add more details of the model configuration including the above motivation. Also we will highlight the uniqueness of our vertical formulation, specifically how it accounts for estuarine circulation (originally only cited - Collins et al 2009.)

   **We have clarified this point beginning on page 4: line 22**

5. *The 2008 paper from Salisbury et al, that is used to back up the scarcity of papers on estuarine carbonate chemistry is outdated by 10 years, and there are indeed some recent papers on this subject that are not mentioned in the manuscript, e.g., Volta et al., 2015 (Hydrol. Earth Sys. Sci.), Cai et al., 2017 (Nat. Comm.) to name a few. There is also older work e.g., Regnier et al., 1997 (Mar Chem.)*

Agreed. We have thoroughly reviewed these suggested studies and will integrate them into the Introduction and Discussion sections of this manuscript.

**We have addressed additional references in the introduction on page 2: lines 9, 11, 16, 17, 20, 28, and 29, and in the discussion on page 16: lines 5 and 33**

6. *while the paper shows that, under some conditions of freshwater influence, the estuarine pH and [aragonite] saturation appears more sensitive, it is not clear why this is so ... procedures to formalize the attribution of processes on pH shifts have been ... recently put in a consistent framework by Hagens and Middelburg, 2016 (Geochim. Cosmochim. Acta)*

We did explore the use of sensitivity (or buffer) factors - e.g., Egleston et al., 2010 (GBC), Soetaert et al., 2007 (Mar. Chem.) - particularly in discussing our sensitivity results along the salinity gradient - e.g., Hu and Cai, 2013 (GRL). Ultimately we decided against using these sensitivity factors since we were trying to communicate the effects of *freshwater* chemistry on properties within a dynamic and productive estuary, which in the endmember mixing case (Section 4.1) are simply caused by the surplus (or deficit) of DIC at the freshwater endmember being mixed into the estuarine zone. We realize that this endmember behavior alone is not new research, but our intention was to put our model sensitivity results in the context of simple mixing. We believe that the endmember mixing analysis supports these results and helps to highlight the important effects of our model sources and sinks within the estuary. We will clarify our intention for the endmember analysis in Section 4.1 and include a brief discussion of why we used that analysis rather than sensitivity factors.

**We have clarified the mechanisms influencing the strength of sensitivity in our model on page 12: lines 23-29. Since changes in the freshwater endmember are affecting conditions in the estuary and buffer factors would describe a change in estuarine sensitivity at given estuarine changes, we opted not to use a buffer factor analysis.**

**Reviewer 2 Response**

(*Reviewer Comments*, Response, Proposed Changes, **Added Changes**)

1. *I must admit that I am conflicted in making a recommendation of this paper. First, this is my favorite subject and I like the approach of a combination of data and modeling (also a combination of numerical model and simple mixing model). However I do not think the combination is successful … If the authors can address these serious issues reasonably, I feel this paper can be a good contribution to our field. Here are my suggestions … if the numerical model study of the Fraser case is new (say it), perhaps expand the model description and limit your discussion to this case, which is essentially what you did but just don't call it global extrapolation.*

   We appreciate the detailed feedback provided by yourself, the other reviewer and the editor, and would enjoy the opportunity to revise the manuscript in accordance. We agree that we have extended some of our analyses beyond their scope and in doing so have unwittingly lost focus in the manuscript. We propose that we scale back to focus our analyses only on the Fraser River. We will still place it in the context of other global rivers (Figure 6) however we will remove the end-member analysis that relied on DIC and TA from other world rivers (Figures 5-d and 7).

   Specifically, we will provide additional detail concerning the Fraser and its estuary, including what is known and less known regarding the drivers of the inorganic carbon cycle. The river is a key driver and yet we have few reliable carbon data in the fresh and brackish waters in the study region. This paucity provides a strong motivation for our analysis. By clarifying this motivation in the text, the key results will be highlighted. We will include the additional references (not all of which were available at the time that this manuscript was submitted) that the reviewers have suggested, where appropriate. We also will define new sensitivity scenarios to include more recent and newly acquired data where possible and reduce (and sometimes remove) the dependence on data in which we have less confidence (such as data collected using outdated methods or river TA with high organic alkalinity uncertainty). We will re-run all the simulations with these new scenarios and produce a new sensitivity summary figure (4), with clarified presentation, to reflect these new results. Finally, we will strengthen the delivery of our main findings and highlight the importance of using the biogeochemical coupled model by refocusing our results (3), discussion (4), and conclusion (5) sections to target the key points below.

   **Key points**

   (a) Responses of estuarine pH and $\Omega_A$ to Fraser River DIC-TA are asynchronous and strongest at opposite ends of the Fraser DIC:TA range.

   (b) Seasonal estuarine productivity reduces estuarine pH sensitivity to river chemistry during summer

   (c) Future Fraser River flow regimes with lower flow in the biologically productive season will favor lower estuarine pH and $\Omega_A$, but the river will dominate a smaller areal region in the estuary.

- We have removed the endmember discussion for other world estuaries beginning on page 14, line 15 and ending on page 15 line 8.
- We have added additional background about the study area starting on page 3: line 32
- We have addressed additional references in the introduction on page 2: lines 9, 11, 16, 17, 20, 28, and 29, and in the discussion on page 16: lines 5 and 33
- We have completely reworked the analysis beginning with new data and analysis (Figs. 2 and S1, Table 2), new model runs and analysis (Figs. 3–7) and strengthened presentation throughout.
- We have clarified our main findings and made them consistent throughout the manuscript in abstract (page 1, lines: 11-19), introduction (page 3: lines 11-14), results (page 11: lines 17-25), discussion (page 12: lines 30-35, page 13: lines 1-12, page 16, lines: 24-34) and conclusions (lines 17-32)

2. *... the paper doesn't present much new data. I believe most new data and the numerical model were published in their two earlier publications (authors really need to say what is new here).*

We appreciate reviewer 2's comment and will revise to clearly lay-out the novelty of this study. This study examines the response of near-surface DIC:TA, pH and aragonite saturation state in our 1-D model (presented originally in Moore-Maley 2016 - detailing model evaluation and basic results/drivers for a *single* river chemistry) across more than 200 year-long runs with different river chemistry scenarios. The results of these runs with varied river chemistry are not published elsewhere. (The run parameters were chosen to simulate our best understanding of seasonal Fraser River DIC and TA, based on previously published total alkalinity observations in and around the river delta, an unpublished mooring pH timeseries in the river delta and finally, limited data mostly with S > 20 from the Fraser estuary - which were published in Ianson et al. 2016.) We will add some more recent new unpublished data from the Fraser estuary (single campaigns in 2014, 2016 and 2017) to further inform new sensitivity scenarios. These data will be highlighted.

We will also shorten and de-emphasize the data methods section (2.2) and move our discussion of organic alkalinity to the study area description (2.1) to clarify that this is a modelling study. We will also scale back our data description in the last paragraph of the introduction to allow the modelling objectives stated there appear more clearly to the reader.

**We have refocused our data analysis to only include estuarine samples for which we have confidence (Fig. 2). We have also condensed our data analysis and merged it with our description of the sensitivity analysis, which comes after the model description on starting on page 6: line 30. This change helps highlight that this is a modelling study.**

3. *... the first sentence in the beginning of the Discussion (p.8, line 18-20) says: "To conceptualize why model estuarine pH is lowest at high $TA_f$" ... this statement is only true in the situation the authors created that is the ratio of river DIC:TA*

*= 1.02 to 1.1 ... it is not "high $TA_f$" but a high (DIC-TA) or DIC:TA in river water that is important here and is the reason behind the phenomenon ... I refer the authors to the paper by Liang et al. 2017 ...*

We agree with the interpretation of reviewer 2 here that the changes in the freshwater DIC-TA (at constant DIC:TA ratio) are responsible for the pH vs S differences between the low and high freshwater TA cases. We also acknowledge that we refer to freshwater TA changes throughout this manuscript without mentioning DIC, which is misleading since we are always in fact manipulating either freshwater DIC:TA or freshwater DIC-TA or both. We will clarify our discussion of river TA scenarios in terms of DIC:TA and DIC-TA.

**We have clarified the roles of DIC:TA and DIC-TA which are discussed on page 12 beginning at line 25, also shown in Fig. 7d**

4. *... in extrapolation of the results, the authors didn't consider temperature effect but this effect can also be significant in controlling carbonate system speciation.*

We plan to limit the generalization of our results in other estuaries and remove Figure 7 and the last 3 paragraphs of Section 4.2.

**We have removed the discussion of endmembers in other estuaries**

5. *If the examples cited by the authors are also true in the Fraser River (e.g., as high as 90% of TA is organic alk), then, how can we believe the $DIC_f$ calculated from the $pH_f$ and $TA_f$?*

We based our freshwater TA endmembers on several data sources, some of which were collected in the Fraser River. By limiting our new endmember scenarios to data collected in the Strait of Georgia where we expect organic alkalinity contributions to be less significant, we can reduce the uncertainty in these scenarios. We will add more recent observations from the Strait of Georgia (see REVISION V above) and remove all TA data collected using outdated methods or in freshwater where organic alkalinity uncertainty is high.

**We have scaled back our data analysis to only include estuarine data. We have mentioned the remaining uncertainty in our analysis on page 7 starting at line 19**

6. *... the extrapolation of the Fraser River DIC:TA ratio globally is just not appropriate ... please fully assess the uncertainty of your assumptions. Here the assumption of $DIC_f$:$TA_f$ > 1.02 probably not just changes the result slightly it perhaps will change the major conclusion derived.*

We have decided to remove the extrapolation to global rivers including the last figure. We will only put the Fraser River in *context* of global rivers (Figure 6 in the original manuscript).

**We have removed this discussion from the manuscript.**

7. *... the paper is poorly prepared and hard to follow (see my detailed reading notes) ... the writing is not transparent to me. So a thorough rewrite with a better readability is also needed.*

The paper will be thoroughly revised (see the "Outline" in our response to the Editor) and we will use your detailed reading notes.

**We have thoroughly reworked this paper beginning from the data, construction of the scenarios, rerunning the model, remaking the figures and rewriting most sections particularly the results, discussion, and conclusions to more clearly guide the reader through our findings and weave our key points in throughout the manuscript.**

8. *Fig. 4 is particularly hard to understand or guess.*

   The revision will include an additional figure with selected (model) timeseries from individual years. We will explain how we go from the time series to points on Figure 4 to clarify where this synopsis figure comes from (see point IX in our response to Editor). We will also clarify the text.

   **We have completely reworked this figure and provided 3 new figures leading into it. We have replaced the envelope and box plots with individual points to show the reader exactly which model runs we are looking at. One of the new lead in plots shows the timeseries of selected runs, illustrating to the reader the seasonal patterns of the model over time so that the salinity averages are less ambiguous.**

9. *... p.9, line 10, it says "This asynchrony arises because the response of estuarine carbonate ion over the large range of river $TA_f$ and $pH_f$ scenarios is more sensitive to changes in total DIC than shifts in the equilibrium point of the carbonate system." First, I don't understand what this sentence really says. Second, it sounds like to suggest that one can change all three parameters ($TA_f$, $pH_f$ and $DIC_f$) at the same time. If here "DIC" is not river DIC but internal estuarine, biologically modified DIC, this is probably true, but the authors didnt say that.*

   We will clarify this sentence. The first phrase is confusing. Briefly, we wish to indicate that the concentration of the carbonate ion is more sensitive to the total amount of DIC (in the estuary) than the balance between the forms of DIC (which varies with pH). Your next point builds on this result.

   **We have rewritten this idea into the new discussion starting on page 12: line 30. We have clearly written the word "estuarine" before any non river value, and subscript $f$ where we refer to freshwater values.**

10. *In the abstract (p.1, line 10), it says "rivers with high DIC and TA produce lower estuarine pH due to an increased estuarine DIC:TA ratio, but higher estuarine $\Omega_A$ because of DIC contributions to the carbonate ion." I like this statement. But I do not really see the result description and an extensive discussion of in the text body. Does this indicate that authors have changed mind a bit on exactly what they want to focus on in this this paper?*

    Thank you. As explained above we will refocus the paper. The difference between the response of pH and $\Omega_A$ is a key point and will be a focus in the revision.

    **We have removed this statement and tried to state this concept in terms of the mechanisms controlling this behavior, see abstract page 1 lines 11-23, results page 11: lines 18-25, discussion beginning on page 12: line 30**

11. *I do not like the implications for future climate. You have speculated too much! (To say high atmospheric $CO_2$ will increase river $pCO_2$ is simply wrong as river $pCO_2$*

*is generally so much higher than the atm-pCO$_2$.) So simplify it and merge it in the Discussion with just a few sentences.*

We will significantly rework our future climate implications section (again, please see the "Outline" in our response to Editor) so that it concerns only the Fraser river and clearly focuses on changes in the Fraser's hydrograph (decreasing freshet) that are anticipated with higher certainty. Potential changes in end-member chemistry will be discussed briefly and with care, clearly detailing the high uncertainty.

Reviewer 2 points out correctly that our text was misleading, and that many rivers are not likely to have increased DIC in future. Some fresh water end-members however may have increased DIC, as they tend to be at atmospheric equilibrium (e.g. glacial melt-water). As reviewer 2 knows, future DIC extrapolations also depend on the concurrent T change (a T increase could leave DIC unchanged even with increased pCO$_2$) leaving significant unknowns and low certainty.

**We have reworked this paragraph to only refer to the Fraser – Strait of Georgia system, and directly address our findings from the sensitivity study. We have also spoken more generally about rising CO$_2$ at either the freshwater or seawater endmember**

12. *Finally your summary is too long and repeats too much of the Discussion.*

    The summary will be focused on the key points list above and we will ensure it is not repetitive.

    **We have rewritten the conclusions to be more concise and tied more closely to our main points**

13. *p.2, Line 15-16, Is this true? I have not seen a river whose TA is NOT flow-dependent. Perhaps, it is because the West paper is about the silicate weathering. Not sure if this is also true for carbonate weathering.*

    We did not intend to indicate that TA in other rivers is not flow-dependent. Our intent was rather to divide strongly flow-dependent from weakly flow dependent. We will rewrite this paragraph for clarity.

    **We have removed this statement since we no longer focus on this type of flow dependence**

14. *p.4, line 15-18, Carbonate Alk is about 50% in the Satilla River Georgia (Cai et al. 1998). Again Cai et al. (1998) paper should be cited as it is the first study of this issue.*

    Noted. We will cite Cai et al. 1998 with respect to organic alkalinity in the new manuscript.

    **We have added a citation for this study on page 4: line 13**

15. *p.4, Line 15-18, this really worries me. If TA data quality is so bad and the organic contribution is so large, how do we know the rest of modeling is correct?*

    We will add more recent discrete observations from the Fraser estuary (point 5 above) and remove all TA data collected using outdated (and even uncertain) methods or in freshwater where organic alkalinity uncertainty is high.

**We have limited our data analysis to estuarine TA data for which we have higher confidence. We continue to use the buoy pH record and acknowledge the remaining uncertainty in our TA extrapolations. We also continue to maintain that these data are simply for guiding our selection of model sensitivity experiments (page 7: lines 22-26). We have also moved our data discussion after the model methods and integrated into the sensitivity methods to further emphasize that the data are not the primary method of the study.**

16. *p.6, Is RMSE = 0.16 pH unit a small uncertainty? It appears quite large to me. Perhaps you need to put it in the context of overall pH change.*

    Good point. We will add the range of pH and also calculate a Willmott score to show the model skill.

    Reviewer 2 is correct that variability in pH is indeed high in the estuary - the new figure with individual year timeseries traces will also add context for the reader.

    **We have rewritten the model evaluation section to give more detail about the model evaluation, clarifying that the large RMSE contains systematic bias, and the non-systematic RMSE is much lower. Given the systematic bias, the model is conservative and overpredicts pH and Omega. See page 6: lines 22-29**

17. *p.7, line 11 says "We define our three constant $TA_f$ scenarios", then line 17 says "We define our three constant $pH_f$ scenarios (Table 2)" (first there is no $pH_f$ in Table 2). I am quite confused if these are related or separated assumptions? From my comment below on Fig. 5 caption, I don't think you need to call them "constant $TA_f$ or $pH_f$", just river endmember scenario 1 to 6 is enough. They have nothing to do with whether $TA_f$ is constant with river discharge; they are just your scenarios.*

    We used a combination of scenarios: so 3 different $pH_f$ and 6 different $TA_f$ for a total of 18 different scenarios, hence 18 boxes on each plot in Table 4. We will completely restyle our presentation our analyses (see response to Editor - VIII) in terms of DIC/TA and DIC-TA and we will assign a run-number, as reviewer 2 suggests, to each model river chemistry scenario. Also, the scenarios themselves will be revised (Editor response – VII).

    **We have summarized our scenarios into Table 2 and refer to them consistently based on that table throughout. We have also improved the figure legends to reflect the layout of that table, especially Figs. 6 and 7. We also refer to the scenarios using consistent language, usually by the actual value (e.g., $TA_f$ = 500 umol/kg), or in the case of $DIC_f$:$TA_f$, "Low, Med and High Carbon"**

18. *p.7, line 20, I do not understand this: "Given the large seasonal temperature change ($> 15\,°C$ increase in summer), a constant $pH_f$ implies a summer $DIC_f$ decrease due to temperature (causing $DIC_f$:$TA_f$ to decrease by about 0.06)". Why does increased temperature lead to $DIC_f$ decrease? Is this just a decreased solubility effect or increased river $pCO_2$ leads to more $CO_2$ degassing? Do you mean that under constant $pH_f$ and $Alk_f$, higher temperature leads to lower $DIC_f$? That is true. But the real question is what is really controlling the $pH_f$ at constant $TA_f$*

*but allowing DIC$_f$ to decrease? You can't just set a certain parameter constant arbitrarily. Anyway you need to explain your assumption. Reading Fig. 5 caption (b), I finally see how you did it. You selected two TA$_f$ values one low and one medium-high. Then at each TA$_f$, you selected three pH$_f$ values and calculated 6 DIC$_f$ values. Then you take these 6 TA$_f$ and DIC$_f$ combinations and mix them with the same seawater (TA$_{sw}$ and DIC$_{sw}$) to generate the 6 mixing lines. I assume at least b and c should be based on these 6 same simulation. (For panel d, it seems you used the variable TA$_f$.) My question is: are these combinations realistic? I now take your 6 combinations and put them into CO2SYS (I assume T = 15 °C and all other acid = 0) and verified/confirmed your DIC$_f$ and DIC$_f$:TA$_f$. No problem except that all DIC$_f$:TA$_f$ ratio > 1.02. This doesn't make sense to me. Perhaps it is true for the low TA$_f$ rivers (but in these river the org-ALK is often large), DIC:TA < 1 definitely occurs in medium and high TA rivers.*

First, we are grateful to reviewer 2 for their meticulous consideration of our scenarios. Reviewer 2 is correct in that our intent was to indicate that *"... at constant pH$_f$ and Alk$_f$, higher temperature leads to lower DIC$_f$ according to CO2SYS"* however, it is clear that our original text was confusing. Our response above to #17 fully clarifies our previous scenarios and more importantly our plans to significantly revamp both the actual river chemistry scenarios and their presentation. (The mismatch in reviewer 2's CO2SYS results and our scenario is due to the fact that the actual T in the Fraser river and in our model is not static - varying from 4-20 °C during the year.)

**We have changed our use of freshwater pH scenarios to using freshwater DIC:TA scenarios. We have also explicitly stated the values for several key carbonate system parameters in fresh water under each of these scenarios in Table 2.**

19. *p.7, line 30-33, It is very hard to understand what the authors try to say after reading this part and Fig. 4 many times. Are DIC:TA (Fig. 4b) or pH (Fig. 4d) averaged over the estuary or what? Very frustrating.*

Again we thank reviewer 2 for their careful review. The revision will include an additional figure with example time series (Editor - IX and response to #8 above). We will explain how we go from the timeseries to points on Figure 4 to clarify where this synopsis figure comes from. The new discussion of the individual traces will clarify the averaging.

**We have completely restructured the presentation of our results, including new figures to lead into the summary figure more gradually, especially Fig. 4 which show timeseries of selected model runs throughout the year. These runs then go on to appear on the summary figure highlighted as red and black stars. We have also included a paragraph in the results dedicated to explaining this figure carefully to the reader. Finally the figure itself is simpler. There are no boxes or salinity envelopes, just a single circle point per run, 216 points in total.**

20. *p.8, the description of Fig. 4 is not very clear. Figure 4 is not clearly labeled. Some supplement instructions are needed. (Like does all gray and white sections in figure 4b, 4c, 4d mean different pH scenario? What does the "5 and 4" mean in figure 4f)?*

A new revision will include an additional figure with example time series. We will explain how we go from the time series to points on Figure 4 to clarify where this synopsis figure comes from. We will complete the labelling.

**See our response to 19 above**

21. *p.8, I am shocked that line 16 moves into Discussion. Now I realized that the entire ms is essentially a model study of pH sensitivity to river discharge and (DIC:TA)$_f$ ratio (the latter is also a function of discharge). Then, I went back to read their two earlier papers. The mode and the field data were already presented there. The authors need to say what is new of the first part of this paper comparing with the earlier papers.*

We see that our original submission failed to put this new work in the context of the original model paper. See our detailed response to #2. Again, our previous papers focused on the estuarine carbon cycle as forced by physical and biological conditions in the estuary. However during that research it became clear that the river chemistry also had a substantial impact. This manuscript looks at that impact. All the model runs are new, as river chemistry was not varied in the previous model paper. We have also collected new data which will be included in the new manuscript.

**We have clarified the nature of this study in multiple places. It is probably best clarified in the model description starting on page 4: line 22 (which now follows directly after the study area description) and at the beginning of the results on page 11: line 1, where there is no longer a presentation of the data. The data discussion is now isolated to section 2.3 where we present the sensitivity experiment methods.**

22. *p.8, line 18-20, see main point.*

Please see our response to the main point (1 above).

23. *p.8, line 28 to p.9, line 8, this is true. In the high river TA estuarine, DIC:TA is decreased slow whereas in low river TA estuarine water, DIC:TA is quickly modified and dominated by the seawater ratio. If this is the only point this paper wants to talk then why presenting the numerical model? Some of the other recent papers also talked about this point. However the assumption of same DIC:TA ratio for high TA$_f$ and low TA$_f$ rivers are likely problematic. Not sure how meaningful is this scenario simulation.*

We are confident that a revision will clearly focus on our key points listed above and that our new scenarios and presentation will facilitate this focus. Also, our addition of model results (which are subject to the dynamics - sources/sinks within the estuary) to the figures presenting the theoretical mixing curves (see Editor response - X) will clearly show the utility of the numerical model.

**We have overlaid model results on top of these theoretical curves and emphasized that the curves are only there to help interpret the model, see page 12 beginning line 13. We then emphasize the importance of model biology in deviating from these curves on page 13 beginning at line 4**

24. *Page 9, Line 8: "Ocean pH and $\Omega_A$ are often assumed to be coupled": Some references may be needed here to support this. I think the word "coupled" and "decoupled"*

*are misused here. In some sense, pH and omega are always coupled. They are just not "coupled" in a simple way as our "intuitions" may suggest. The simple case here is that $[Ca^{2+}]$ increased as salinity increases, but that has no direct effect on pH. There are other more subtle factors or processes influencing pH and omega differently. However I do not think we can simply call that as "decoupled". You can find a better name.*

Noted. In the new manuscript we will not use the word "coupled" but instead note that non-experts often assume lower pH implies lower $\Omega_A$.

**We no longer use the word "coupled" and now use either "asynchonous" or "trend reversal" See page 12 beginning at line 30.**

25. *p.9, line 12-14 is not clear.*

    Noted. In the new manuscript we will expand and clarify as this is a key point of the paper.

26. *p. 10, I like the discussion on seasonality, but it is based on physics ($TA_f$ and discharge only).*

    We will remove this section as we will focus on the Fraser River.

27. *p.10, line 14-16, true that high river $TA_f$ systems like the Mississippi provide a strong buffer effect and its delta-pH shouldn't change as much as that in low $TA_f$ systems during mixing. However biological production could raise pH to a very high value in the Mississippi.*

    Good point. The new manuscript will focus on the Fraser River.

    **We have removed our discussion of other estuaries**

28. *Fig. 6c and d, Note there are two arms of DIC:TA ratio to TA with a minimum at seawater TA. There is nothing magic here but the authors should mention the reason. The left reflects the mixing between the generally high ratio in river with a low ratio in seawater. The right arm reflects the mixing of a few very high TA rivers (with TA higher than the seawater) with seawater.*

    As we will focus the new manuscript on the Fraser River, we will remove Figure 6d.

    **We have removed panel d**

29. *p.11, line 1 and 6 are not consistent. Warming allow more $CO_2$ degassing and decreases river DIC. Yes, I agree. But increased atmospheric $CO_2$ probably won't increase DIC as river DIC is so much higher than the atm-pCO_2. If there is any increase it likely increases TA equally (through increase of weathering rate).*

    In the new manuscript we will only consider the Fraser River and future climate changes we are sure about, for example, changes in timing of the freshet.

    **We have reworded the discussion more broadly to consider general increases in the dissolved $CO_2$ of either endmember, see page 17: line 30**

30. *p.10-11, I think this section is very speculative and should be deleted or combined into earlier discussion with short sentences. These speculations do not help, e.g.,*

*we do not know which competing factor will dominate and if river DIC will increase. Overall as the authors agree that these effects are rather small comparing with eutrophication induced surface biological production and subsurface respiration induced pH changes.*

In the new manuscript we will only consider future climate changes we are sure about, for example, changes in timing of the freshet.

**We now only consider changing physical flow regimes and increasing dissolved carbon. We can tie both of these processes directly to our key points, and they are clearly illustrated in our new summary figure, Fig. 6. Fig. 6 also shows the magnitude of sensitivity is not insignificant relative to processes like eutrophication.**

31. *p.11-12, I generally do not like long summary, which essentially does more repeating discussion.*

The summary will be focused on the key points list above and we will ensure it is not repetitive.

**We have completely rewritten the conclusions to more concisely address our key points**

32. *Finally, regarding pH scale. I do not understand why the authors switches between NBS and total scales. I'll stick with one and note there is big uncertainty in either one when salinity is extreme (that is $pH_T$ doesn't work for river water and $pH_{NBS}$ doesn't work fully for seawater). Also, pH was given as in "NIST units". This is not the right way we marine chemists will say. It should be in "NIST scale" (I would just call it "in NBS scale"). When saying a pH change then you can say "a change of 0.xx pH units". There is not such a name called NIST or NBS or total units. It is scale!*

We will use only the total scale for pH and convert all freshwater pH to total scale before presenting them in the manuscript. We will not call them units.

**Use use the total scale exclusively in our revision.**

33. *Since the ms is an open access discussion paper, I also asked a colleague who knows statistics better than me to read it. Below is her comments. I have read these and generally agreed.*

*Page 6 Line 10: the positive bias, root-mean squared error of the model output is 0.16 for pH and 0.51 for the saturation state of aragonite. The authors claim that these errors are sufficiently small to support the model use for the process studied in the paper. I am not sure about this claim. The error of 0.51 looks big enough to me from my understanding of acidification impacts. The model may be good in reproducing the physical field and biology bloom as the authors stated here.*

*A regression with $R^2$ of 0.1 without a P value is impossible for readers to judge whether the regression is significant or not (Fig. 3b). If the regression is not significant, TA is not flow dependent, and then the model can't use this relation to derive a scenario. For the regression-based scenarios, since data vary greatly, uncertainties associated with these regressions should be provided and transferred to the model outputs. Without knowing the uncertainties, considering the error of the*

*model output of the saturation state of aragonite, 0.51, not so small, it is hard to evaluate the duration of $\Omega_A < 1$ in any scenario. The authors may specify how low $\Omega_A$ is in these scenarios.*

Figure 3b and all TA data collected using outdated methods or in freshwater where organic alkalinity uncertainty is high will be removed. We will define new sensitivity scenarios to include more recent and newly acquired data where possible and reduce (and sometimes remove) the dependence on data in which we have less confidence (such as data collected using outdated methods or river TA with high organic alkalinity uncertainty). We will also base our flow-dependent scenarios based on theoretical weathering curves spanning the range of observations rather than statistical fits. We will re-run all the simulations with these new scenarios.

The model $\Omega_A$ uncertainty of 0.51 is large as reviewer 2 mentions, but also positively biased and thus does not overestimate the severity of aragonite undersaturation. We will reword our discussion of the model evaluation to emphasize the bias rather than the uncertainty as our motivation to go forward using the model in this study.

- **We have revised the model evaluation section to give more detail. See our response to 16 above.**

- **We have only proposed a single flow dependent scenario for our revised sensitivity analysis. We do not have enough data to demonstrate statistical significance, but, provided our assumption that TA is approximately conservative across the lower salinity range where we don't have data, then we have high confidence in our higher salinity data since we we're involved in its collection, and that confidence translates to the errorbars shown in Fig. 2.**

[revised manuscript text omitted]

---

## Author Response (AR2)

**Reviewer 2 Response – second round of review**

(*Reviewer Comments*, Response, Changes)
All line numbers refer to marked version

*This paper has examined the sensitivity of $\Omega_{arag}$ and pH in the Strait of Georgia to Fraser River carbonate chemistry by summarizing the results of a coupled biogeochemical model across 12 hydrological cycles and 18 freshwater TA and DIC:TA combinations. The simulations and discussion are of interest to the current research. I agree that the revised manuscript is much improved. I generally support the publication after additional improvement of writing.*

We are grateful that Reviewer 2 has agreed to review our manuscript a second time and we are happy to hear that they find the document much improved. We have included our responses to these additional revision suggestions below.

*Below are my two main concerns.*

1. *While I can understand the abstract and summary very well, I feel some of the presentations on the Results and Discussion are still quite hard to follow. I want to warn the authors (and the editor) that other readers may have similar issues in assimilating the ideas from this paper.*

   Upon reading through the Results and Discussion again, we identified the following areas for improvement.

   - Our description of the trends and variability in Fig. 6 is not clearly linked to the horizontal and vertical axes
   - Our presentation of the endmember mixing curves and their similarities to the model results in the first paragraph of 4.1 is cumbersome
   - Paragraph 3 of 3.1 (discussion of flow-dependent TAf) feels isolated and out of place
   - Some sentences run on for several lines and can be shortened
   - Some statements are repetitive and unnecessary. These statements are particularly troublesome when they contain symbols and number values.

   We've made the following specific revisions to the results (Sect. 3.1 and 3.2) and discussion (Sect. 4.1) to address the issues stated above.

   - We clarified the effect of freshwater flux in Fig. 6 as the variability along the vertical axis (Page 8, Line 2)
   - We've broken some of the longer sentences into smaller parts (Page 8, Lines 5-9, 31-34; Page 10, Lines 7-10)
   - We removed repetitive statements (Page 9, Lines 5-8)
   - We more clearly state the purpose of the flow-dependent results paragraph (Page 8, Line 12)

We've also overhauled the first paragraph of the discussion (Sect. 4.1) to more clearly state

- what the endmember mixing curves represent and why they were done (Page 9, Lines 22-26)
- how the mixing curves and the model results are similar (Page 9, 26-33)
- the paragraph objectives in plain language rather than symbols

We've also made small changes for clarity throughout Sections 3 and 4.

2. *As stated in the paper (e.g., abstract and summary), the Fraser River pH and $\Omega$ are strongly controlled by biological production during the production seasons and, even during the maximum river flow summertime, estuarine sensitivity to river chemistry is greatly reduced. Then, I am not totally convinced that such a lengthy discussion on this aspect is fully justified. Maybe some additional justifications are needed on that the sensitivity to river chemistry is still an important consideration.*

We have highlighted the significant effects of river chemistry in the presence of biology throughout the manuscript and they are summarized in Figures 6 and 7. Thus to address this particular concern we have decided to scale back the emphasis that we place on the importance of estuarine biology in mitigating these effects.

We have made the following changes.

- Abstract: Page 1, Line 16. "**Significantly** modifying these negative impacts ..." remove 'significantly'
- Discussion (4.1): Page 10, Line 19. "... the importance of biology and gas exchange in **mitigating** unfavorable carbonate chemistry ..." change 'mitigating' to 'reducing'
- Discussion (4.1): Page 10, Line 21. "... estuarine carbon decreases **dramatically** during the spring phytoplankton bloom ..." change 'dramatically' to 'significantly'
- Conclusion: Page 12, Line 27. "Once the spring phytoplankton bloom occurs ... both estuarine pH and $\Omega_A$ increase **markedly** away from the physical mixing line." remove 'markedly'

*Below are some minor issues I spotted while reading:*

1. *p.1, Line 9, while the carbonate portion of the (higher) DIC causes an increase in estuarine A... is confusion. Simply say while the resulting higher carbonate ion concentration causes an increase in estuarine A... is clear and sufficient.*

Done at Page 1, Line 12.

2. *p.2, line 15, the citation Cai et al. 1998 is probably wrong and the correct or more appropriate one would be Cai et al. 2008.*

*Cai, W.-J., Guo, X., Chen, C.T. A., Dai, M., Zhang, L., Zhai, W., Lohrenz, S.E., Yin, K., Harrison, P. J. and Wang, Y. 2008. A comparative overview of weathering intensity and $HCO_3^-$ flux of the world's major rivers with emphasis on*

*the Changjiang, Huanghe, Zhujiang (Pearl) and Mississippi Rivers. Continental Shelf Research 28:1538-1549, doi:10.1016/j.csr.2007.10.014.*

Done at Page 2, Line 16.

3. *Bottom of p.6, eqn 1. I assume the flow dependent river endmember TA can be smaller and greater than the TA0. But just looking at eqn 1, that is not so clear to me. I guess it is in the dQ/dt term, which can be negative. please add a short note (I now see that a reference to fig. 2c is all it needs).*

We've added the following sentence at the top of page 7:

[revised manuscript text omitted]